



# Global evaluation of RTTOV coefficients for early satellites sensors

Bruna Barbosa Silveira[1,*], Emma Catherine Turner[2], and Jérôme Vidot[1]

[1]CNRM, Meteo-France/CNRS, Université de Toulouse, Lannion, France
[2]Met Office, Fitzroy Road, Exeter, EX1 3PB, UK
[*]Current Affiliation: Laboratoire de Météorologie Dynamique (LMD/IPSL), École polytechnique, Institut Polytechnique de Paris, Sorbonne Université, École normale supérieure, IPSL Research University, CNRS, École des Ponts, F-91128 Palaiseau, France

**Correspondence:** Barbosa Silveira (brunabs.silveira@gmail.com)

**Abstract.** RTTOV coefficients are evaluated using a large, independent dataset of 25,000 atmospheric model profiles as a robust test of the diverse 83 training profiles typically used. The study is carried out for nine historic satellite instruments: IRIS-D, SIRS-B, MRIR and HRIR from the infrared part of the spectrum, and MSU, SSM/I, SSM/T2, SMMR and SSMI/S from the microwave. Simulated channel brightness temperatures show similar statistics between both the independent and the 83 profile datasets, confirming that it is acceptable to validate the RTTOV coefficients with the same profiles used to generate the coefficients. Differences between RTTOV and the line-by-line models are highest in water vapour channels, where mean values can reach up to 0.4 ±0.2 K for the infrared, and 0.04 ±0.13 K for the microwave. Examination of the latitudinal dependence of the bias reveals different patterns of variability between similar channels on different instruments, such as 679 cm$^{-1}$ on both IRIS-D and SIRS-B, showing the importance of the specification of the ISRF. Maximum differences up to several kelvin are associated with extremely non-typical profiles, such as those in polar or very hot regions.

## 1 Introduction

The fast radiative transfer model RTTOV (Radiative Transfer code for TOVs (TIROS Operational Vertical sounder)), Saunders et al. (2018), is used as the observational operator that assimilates satellite measurements in multiple Numerical Weather Prediction (NWP) models, see Eyre et al. (2022), enables the retrieval of atmospheric or surface parameters (Merchant et al., 2019) or the simulation of satellite imagery from NWP models, and is also widely used across the world as a stand alone model for scientific research applications, for example Chen and Bennartz (2020). The 'fast' nature of RTTOV is attributed to the linear regression methods at its core, which combines pre-trained satellite coefficients with various combinations of predictors for each atmospheric constituent, in place of the full line-by-line atmospheric absorption calculation. Though highly accurate, this parameterisation introduces a small error in the calculated radiance, which can be quantified by comparison with the equivalent results obtained from a high-resolution line-by-line (LBL) model (Saunders et al., 2007).

The evaluation of RTTOV's performance presented in this study was carried out as one of the radiative transfer objectives for a project within the European Union's Copernicus Climate Change Service[1] (C3S), entitled 'C3S 311c: Support for climate

---

[1]https://www.copernicus.eu/en/copernicus-services/climate-change



reanalysis including satellite data rescue'. C3S is a program implemented by the European Center for Medium-Range Weather Forecasts (ECMWF) that combines observations of the climate system with the latest science to develop authoritative, quality-assured information about the past, current and future states of the climate in Europe and worldwide. Recent advances in satellite data rescue for early satellite instruments are presented in Poli et al. (2017).

The first phase of the C3S 311c project comprised several concurrent data rescue studies that were carried out between 2019 and 2021, two of which were satellite based. The aims were to retrieve/rescue and reprocess historic infrared and microwave (MW) meteorological satellite observations from the 1970's and 1980's, primarily for inclusion in the ECMWF ERA6 re-analysis, the follow-on to ERA5 (Hersbach et al., 2020), which is due to be released in 2024. In these reanalyses, RTTOV is employed to simulate all satellite observations. An overview of all RTTOV objectives within C3S 311c project is given in Vidot et al. (2021).

Current validation of RTTOV coefficients for clear-sky simulations is based on the comparison between LBL simulations of the standard 83 training profiles used for coefficient generation, versus the same results from RTTOV. This validation is done for all instruments simulated by RTTOV and statistical plot and data can be found on the NWP Satellite Application Facility (NW-PSAF) website[2]. A more robust validation can be obtained by employing a larger independent profile dataset. The objective of this study is to provide further validation of the RTTOV coefficients by using a much larger independent dataset composed of 25,000 globally distributed profiles, selected from one year of the IFS (Integrated Forecast System) NWP model and provided by NWPSAF. The validation is studied for the high priority infrared instruments identified by the C3S 311c project, which are: InfraRed Interferometer Spectrometer - D (IRIS-D), Satellite Infrared Spectrometer A and B (SIRS-A and SIRS-B), Medium Resolution Infrared Radiometer (MRIR) and High Resolution Infrared Radiometer (HRIR) and for the microwave instruments: Microwave Sounding Unit (MSU), Special Sensor Microwave Imager (SSM/I), Special Sensor Microwave water vapor profile (SSM/T-2) , Scanning Multichannel Microwave Radiometer (SMMR) and Special Sensor Microwave Imager/Sounde (SSMI/S) (Table 1).

The 25,000 independent profiles and the 83 training profiles will be presented in section 2. Section 3 describes the LBL models used for the infrared and the microwave, line-by-line radiative transfer model (LBLTRM) and Advanced Microwave Sounding Unit Transmittance Model (AMSUTRAN), respectively, and the RTTOV versions used for the simulations. The satellite instruments and the 25,000 independent profiles evaluated against the training profiles using the mean, standard deviation and maximum differences between LBL and RTTOV simulations for both dataset will be presented in Section 4. Analysis of the global position of each of the independent profiles, and the spatial and latitudinal distribution of differences will also be shown for selected channels. The fifth section summarises the main results and draws conclusions from the previous sections.

## 2 Atmospheric profiles

The diverse profile training dataset contains 83 profiles for six molecules (water vapour ($H_2O$), ozone ($O_3$), carbon dioxide ($CO_2$), methane ($CH_4$), nitrous oxide ($N_2O$) and carbon oxide (CO)) and one standard profile, mostly from the US76 standard

---

[2]https://nwp-saf.eumetsat.int/site/software/rttov/download/coefficients/comparison-with-lbl-simulations/





atmosphere database on Extension to the Standard Atmosphere (1976), is used for the other 22 molecules and Chlorofluoro-carbons (CFCs), though not every molecule is included for every instruments, depending on its spectral absorption coverage. The atmospheric profiles for the first six molecules were selected from a large database originally on 91 levels generated by the experimental suite (cycle 30R2) of the ECMWF forecasting system, as described in Chevallier et al. (2006). The 81, 82 and 83 profiles are the minimum, maximum and mean, respectively, of the initial database. We refer the reader to the RTTOV science

and validation reports for a complete description of the training dataset, listed in Saunders et al. (2017) .

The larger independent set of atmospheric profiles used is described in Eresmaa and McNally (2014). Only values for water vapour, temperature and ozone profiles were used in the present evaluation. This dataset includes 25,000 profiles divided into five subsets each with 5,000 profiles. The profile is split into 137 levels from the surface to 0.01 hPa, which is the resolution currently used by the Integrated Forecasting System (IFS) developed at ECMWF. The dataset is selected from the short-

range IFS forecast over one year and is available from the NWPSAF website[1]. Each of the 5,000 profile subsets represents maximum diversity of one of five different variables: temperature (t), specific humidity (q), ozone (o3), cloud condensate (ccol) or precipitation (rcol).

Figure 1 shows the temperature profiles (grey lines) for the 25,000 profiles and 83 training profiles (Fig. 1a and 1b, respectively). The mean profile for the full 25,000 dataset (green lines) and for the 83 profiles (red lines) are also plotted on

each panel. The vertical distributions of the maximum and minimum values are similar between the subsets and the training profiles, however, the mean stratopause height around 1 hPa is noticeably higher in the training set. This is probably due to the different vertical resolutions of the initial data from the ECMWF model, where the 137 levels of the larger dataset will better resolve the upper atmosphere than the 91 levels that comprise the training profiles. The maximum (minimum) temperature of the independent profiles is 319.39 K (160.29 K) and for the training profiles it is 318.25 K (159.61 K).

Figures 1c and 1d also show the water vapour profiles of the 25,000 profiles subsets and the training profiles, respectively. In the upper troposphere the profiles from the independent profiles generally have less water vapour than in the training profiles. The water vapour mean profiles (green and red lines) are similar. The minimum values of the two datasets are the same (0.016 ppmv) and the maximum values are 41.881 ppmv in the independent profiles and 42,868 ppmv in the training profiles.

The vertical distributions of ozone profiles are shown in Fig. 1e and 1f. The training profiles consist of less ozone in the

stratosphere when compared to the independent dataset, which can be seen in the mean values. The mean ozone value around the peak level at 10 hPa is smaller in the training profiles relative to the independent profiles (7 ppmv versus 8 ppmv) and at this altitude the ozone variability is larger in the training profiles (Fig. 1f). The small values of ozone in the training profiles could be due to a profile located at the ozone hole. Ozone values vary between 0 and 10.65 ppmv in both datasets.

The ozone distribution for all five subsets contains some profiles with a second peak of ozone above 1 hPa (only the ozone

subset is shown in the Fig. 2), whereas this behaviour is not present in the training profiles. These profiles are located mainly in the polar regions. Figure 2a shows the spatial distribution of the profiles with a second peak taken from the ozone subset (the same behaviour is observed in the double peak profiles from the other four subsets) and Fig. 2b shows their vertical distribution.

---

[1]https://www.nwpsaf.eu/site/software/atmospheric-profile-data/



The profiles were selected when the ozone content exceeds 3ppmv above 0.9 hPa. In these profiles the ozone concentrations are lower than average in the troposphere and mesosphere.

## 3 Line-by-line models and RTTOV

### 3.1 RTTOV setup

The RTTOV simulations were processed with the same profiles used for the LBL simulations where any profiles that failed the LBL models were excluded. Independent profiles were interpolated from the original 137 model levels to either 101 or 54 pressure levels, depending on the instrument. For hyperspectral instruments 101 levels is used as the full vertical stratification of the atmosphere is resolved, but 54 levels is sufficient for most narrowband instruments. For the infrared instruments the version 7 predictor RTTOV coefficients (with 101 levels for IRIS-D and 54 levels for other instruments) were used because there is no variation in the carbon dioxide ($CO_2$) profile. As with the LBLRTM simulations, the $CO_2$ profile used in the RTTOV simulations was the mean training profile (profile number 83). The $CO_2$ value in profile 83 is around 400 ppmv, as described in Saunders et al. (2017). The infrared simulations were made only at nadir and with surface emissivities equal to 1. The RTTOV setup for the microwave instruments was the same as for the infrared instruments except for the satellite angle. For the microwave simulations, the standard six satellite zenith angles that vary between 0.0° to 63.6° were used, which equates to secant values of: 1.0, 1.25, 1.5, 1.75, 2.0 and 2.25. All microwave RTTOV simulations use version 7 predictors, and 54 (as opposed to 101) levels were sufficiently accurate for this analysis.

### 3.2 Line-by-line radiative transfer model setup

#### 3.2.1 Infrared instruments

The LBL simulation of the radiances at the top of atmosphere (TOA) were performed with the Line-by-Line Radiative Transfer Model (LBLRTM) version 12.2 (Clough et al., 2005). LBLRTM v12.2 combines the Atmospheric and Environmental Research (AER) v3.2 molecular line database and the MT-CKD absorption continuum version 2.5.2. The AER v3.2 molecular database is based on High-resolution Transmission molecular Absorption database (HITRAN) 2008 (Rothman et al., 2009) with many improvements concerning positions and intensities of some molecules and line-mixing effects[2]. The 101 (54) level profiles were used to simulate the hyperspectral (other) sensors. As previously mentioned, the independent profiles only contain temperature, water vapour and ozone information, and profiles relating to the molecules: $CO_2$, $CH_4$, $N_2O$ and CO are the mean training profile set (profile 83), and one US76 standard profile for other the 22 other molecules. The LBLRTM simulations were performed in one continuous band from $75\,\mathrm{cm^{-1}}$ to $3325\,\mathrm{cm^{-1}}$ with a spectral resolution of $0.001\,\mathrm{cm^{-1}}$. The LBL simulations were then convolved by the Instrument Spectral Response Functions (ISRF) of each sensor. To perform the convolution from radiances to instrument channels the ISRF used to generate the RTTOV coefficients were applied. The ISRF are available at

---

[2]see a full description for each molecule at https://github.com/AER-RC/LBLRTM/wiki/What's-New

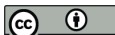



the NWPSAF website[3]. It was found that a small percentage of the independent profiles could not be simulated by LBLRTM. This could be explained by some inconsistencies in the profile arising from the vertical interpolation. In the simulations with 101 levels, 40 profiles failed (0.16%). In the simulations with 54 levels, 55 profiles failed (0.22%).

### 3.2.2 Microwave instruments

The line-by-line simulations for the microwave instruments were performed with AMSUTRAN (Turner et al., 2019), a line-by-line code dedicated to producing channel-averaged transmittances for microwave and sub-millimetre instruments, maintained by the NWP SAF. The spectroscopy is based on Liebe (1989) with modifications made to key lines over time, such as the broadening parameters of the 22.235 and 183.31 GHz water vapour lines. Other major changes include updating all oxygen parameters with those provided by Tretyakov et al. (2005) and adding 35 of the strongest ozone lines below 300 GHz with parameters from HITRAN 2000 (Rothman et al., 2003).

The 25,000 profiles were interpolated to 54 levels before being ingested into the line-by-line code. The only gases included in the calculation are water vapour, oxygen, nitrogen (continuum only) and ozone, where the latter three are combined into a single mixed gases transmittance profile. AMSUTRAN produces instrument radiance by performing an 'on-the-fly' calculation on a fine spectral grid over the bandwidth of each channel, the resolution of which is pre-determined based on the features of the spectrum, i.e. close proximity to a sharp oxygen line necessitates a higher resolution than a channel in a window region, for example. Channel resolution for these five microwave instruments ranges between 0.005 and 50 MHz. The mean of transmittance over the spectral grid gives one profile for each channel for both water vapour and the mixed gases. No spectral response function is applied as these were found not to be available for these historic microwave instruments, so each channel assumes a top-hat/boxcar shape. This means that the choice of satellite for each series of instruments is immaterial, as they will all be the same.

The microwave simulations are less computationally intensive than the infrared ones, hence it was possible to look at all six standard satellite zenith angles included in the RTTOV coefficients, as well as the nominal nadir view. Of all five microwave instruments only two, MSU and SSM/T-2, are cross-track scanners so for these two the full range of angles are presented. The remaining three are conical scanners where the zenith angle is fixed, however all six angles are still included in the coefficients. As for infrared the microwave surface emissivity for all simulations is set to 1. Radiance and brightness temperature are calculated using a linear-in-tau approximation with the transmittance profile, see Berk et al. (1998). The same approximation was applied to the infrared too.

## 4 Independent profile dataset versus training profile dataset

A description of each of the sensors used in this study follows. The evaluation was performed separately for each subset of 5,000 profiles. However, the statistics were almost the same for each subset and for the ensemble of 25,000 profiles. For

---

[3]https://www.nwpsaf.eu/site/software/rttov/download/coefficients/spectral-response-functions/





this reason, only the statistics for the 25,000 profiles (together) are shown. Note that maximum difference can be positive or negative with respect to the order of the subtraction between datasets.

## 4.1 Statistical Evaluation

### 4.1.1 IRIS-D

The InfraRed Interferometer Spectrometer (IRIS-D) was a hyper-spectral IR sensor, which had 862 channels (from 400.47 $cm^{-1}$ to 1597.71 $cm^{-1}$), 94 km of spatial resolution at nadir and flew on Nimbus 4 (Hanel et al., 1971). Figure 3 shows the TOA Brightness Temperature (BT) as mean (a), standard deviation (b) and maximum value (c) of the differences between RTTOV and LBLRTM for all IRIS-D channels. Overall, the mean differences between the independent profiles (blue lines) and the training profiles (red lines) are very similar, and vary between -0.238 and 0.247 K. The differences between the two datasets are more evident in the $CO_2$ band (between 600 and 800 $cm^{-1}$) and in the ozone band (near 1000 $cm^{-1}$). A similar behaviour is found in the standard deviation of differences, which varies between 0.006 K to 0.235 K. The highest standard deviation, and mean differences between datasets, are in the H2O channels below 600 $cm^{-1}$ and above 1300 $cm^{-1}$. The variability of water vapour profiles in the independent profiles is greater than in the training profiles, which could explain why they are higher. Conversely, the variability of ozone profiles from the training profiles is higher in the ozone peak, which could explain the higher values of standard deviation of mean differences in the ozone band. The statistics from the independent profiles have higher maximum values than the ones from the training profiles (up to 6 K in some channels but mostly below 2 K against 0.5 K in the training profiles) which is to be expected because they have more variability. The standard deviation of the differences present a small value when compare against the instrument noise. The instrument noise is near 0.5 K between 600 and 1200 $cm^{-1}$, it decreases from 4 to 0.5 k between 400 and 600 $cm^{-1}$ and it increase form 0.5 up to 3.5 K above 1200 $cm^{-1}$.

### 4.1.2 SIRS-A and SIRS-B

The Satellite Infrared Spectrometer A (SIRS-A) was an infrared sensor which had eight channels (668.7 $cm^{-1}$ to 899 $cm^{-1}$) and flew on Nimbus 3 (Wark, 1970). SIRS-B was an infrared sensor which had 14 channels (280 $cm^{-1}$ to 899 $cm^{-1}$), and increase of six relative to SIRS-A, and flew on Nimbus 4 (Hanel et al., 1972). The spatial resolution was 220 km at nadir for both sensors.

Figure 4 shows the mean (a), standard deviation (b) and maximum value (c) of the differences between RTTOV and LBLRTM simulations for each channel for SIRS-B. For SIRS-B, statistics for the independent profiles and the training profiles have similar values. The mean differences for the training profiles (red bars) are larger than the independent profiles (blue bars) only in the channels centred at 14.95 $\mu$m and 22.91 $\mu$m, and for all other channels the reverse is found. The statistics for the training dataset are very good for channels below 15 $\mu$m as compared with the independent dataset but overall the mean differences are between -0.05 and 0.2 K for all channels and for both datasets. The standard deviations are slightly larger in the independent profiles but the two datasets have similar values, and for both datasets the standard deviation increases with





wavelength. As for the IRIS-D channels, the maximum values for the SIRS-B are higher for the independent profiles, with values up to 3.5 K against 0.5 K for the training dataset.

The first eight SIRS-B channels show statistics similar to the SIRS-A channels. The mean differences for the training profiles (red bars) are larger than the independent profiles (blue bars) only in the channels centred at 14 $\mu$m and 14.95 $\mu$m, and for all other channels the reverse is found. The statistics for the training dataset are very good for channels below 14.94 $\mu$m as compared with the independent dataset but overall the mean differences are between -0.05 and 0.1 K for all channels and for both datasets (figure not shown). The standard deviations are slightly larger in the independent profiles but the two datasets have

similar values, and for both datasets the standard deviation increases with wavelength, which is more evident in the independent profiles. As for the IRIS-D channels, the maximum values for the SIRS-A are higher for the independent profiles, with values up to 1 K against 0.2 K for the training dataset. These statistics of SIRS-A were also compare against the instrument noise. The channel 14 $\mu$m presents the highest value (0.7 K) and in the other channels the noise varies between 0.1 and 0.25 K which are higher when compared against the standard deviation of the differences (figure not shown).

Other two sensors were also analysed. The High Resolution Infrared Radiometer (HRIR) which had only one channel (3.76 $\mu$m), spatial resolution of 8 km and flew on Nimbus 1, 2 and 3. The statistics are very similar for both profile datasets (figure not shown). The other sensor is the Medium Resolution Infrared Radiometer (MRIR), this sensor had four channels (centred between 6.62 $\mu$m and 17.06 $\mu$m), flew on Nimbus 2 and 3 and presented a spectral resolution of 55 and 45 km. The statistics of this sensor is similar to the other two sensors, except in the channel centred at 17.06 $\mu$m which presents the highest mean

differences, standard deviation and maximum value. The main cause probably being the fact that this channel has a very large bandwidth (between 5 and 30 $\mu$m).

### 4.1.3   MSU

The Microwave Sounding Unit (MSU) was a Dicke-type cross-track radiometer with a 110 km footprint (at nadir) that successfully flew on nine NOAA satellites between TIROS-N and NOAA-14 (not NOAA-13), Spencer and Christy (1990). It had

four temperature sounding double passband channels in the 50–60 GHz oxygen region, each with a bandwidth of 200 MHz and NE$\Delta$T (Noise Equivalent $\Delta$ Temperature) of 0.3 K.

    Temperature channels tend to perform well in RTTOV compared with the underlying line-by-line model and this instrument is no exception (Fig. 5). In general the differences between the two profile datasets are small, with a slight decrease in the mean and standard deviation but increase in the maximum value when using the larger dataset – all of which would be expected. The

maximum difference in any channel is below 0.15 K.

    Variation with satellite zenith angle varies depending on channel, but the pattern of variation is similar between the two datasets. Channel 1 is more like a window channel than the other three and shows more dependency on SZA in general. Mean biases reduce and then become negative with increasing angle, giving an overall six angle mean of near zero. Standard deviations increase slightly and so does the value of maximum difference, with bigger values in the larger dataset, up to 0.5

K. The other three channels show relatively little angular dependence in the mean, a slight increase (or decrease in the case of channel 3) in the standard deviation, and little dependence in the maximum differences, apart from channel 4 in the larger





dataset, where values increase with satellite zenith angle up to 0.15 K. Particularly in the standard deviations, there is relatively little difference in statistics when using the nadir view only or all six satellite zenith angles.

### 4.1.4 SSM/T-2

The Special Sensor Microwave water vapor profiler (SSM/T-2) was a total power cross-track radiometer onboard four of the Defense Meteorological Satellite Program (DMSP) satellites between F11 to F15 (not F13), Galin (1993). It had five double passband channels, the first two centred at 91.6 and 150 GHz, respectively, being effectively window channels, and the remaining three sensing incrementally closer to the 183.31 GHz water vapour line. Footprint sizes ranged from 88 km for the 91.6 GHz channels to 48 km for all of the water vapour channels, at nadir. All apart from the highest peaking channel 5 (183.31

$\pm$ 1.0 GHz) had an NE$\Delta$T of 0.6 K, whereas channel 5 had a value of 0.8 K.

For SSM/T-2 there is an approximate 10-fold increase in all difference statistics in comparison to MSU, which is to be expected as water vapour is more difficult to predict than dry air, due to its increased variability (Fig. 6). The pattern of mean statistics look quite similar between datasets. As with MSU, the maximum differences increase in the 25,000 profile set, however, the standard deviation increases instead of decreases. As SSM/T-2 is a humidity sensitive instrument this is likely due

to the broader range of water vapour profiles in a far larger dataset relative to the possible dry air profiles. Channel 3, which is the channel closest to the centre of 183.31 GHz line shows the largest increase in standard deviation and maximum difference, with the maximum changing from 0.2 to nearly -2 K.

There is a similar variation of satellite zenith angle dependency in the mean bias of all channels, however the standard deviation of the bias is relatively independent of angle. Both nadir and the total angular mean are very close in value indicating

one or the other could be used in bias correction schemes. Maximum differences become larger (in absolute value) with increasing zenith angle, almost doubling for most channels and profile sets.

### 4.1.5 SSM/I

The Special Sensor Microwave Imager (SSM/I) was a conical scanner that flew onboard seven of the DMSP satellites from F8 – F15 (not F9),Hollinger et al. (1990). It comprised seven double passband channels with four frequencies, three of which

had vertical and horizontal polarisation, and the remaining channel vertically polarised and centred directly on the 22.235 GHz water vapour line. The other frequencies are in semi-window regions centred at 19.35, 37 and 85 GHz. Nadir footprint sizes vary between around 43 km and 13 km depending on channel. NE$\Delta$T values vary between 0.37 and 0.73 K.

There are similarities in the difference patterns to SSM/T-2 (not shown) as they are both window/water vapour sensing instruments and there are slightly larger difference statistics in the 25,000 profile dataset for all channels, however, the magnitudes

for SSM/I are about 10-fold less than those for SSM/T-2 as they are at lower frequencies. Channel 3 at 22.235 GHz shows some similarities to the 183.31 GHz channels on SSM/T-2, however, the mean differences in the 25,000 profile dataset are larger in the 85.5 GHz channels. The maximum difference is -0.3 K in channel 3. When all six zenith angles are included mean differences reduce quite significantly for both profile datasets, however, standard deviations increase. The maximum difference increases to -0.6 K in channel 3 though the other channels only reduce slightly.





### 4.1.6 SMMR

The Scanning Multichannel Microwave Radiometer (SMMR) was an early conical scanning instrument that flew on Nimbus 7 (Gloersen and Barath, 1977), and on a demonstrator mission on SeaSat but this is not considered in this project. SMMR comprises ten single passband channels with five frequencies - one channel each for either vertical or horizontal polarisation. The frequencies are all between 6.6 and 37 GHz and are primarily window channels with some influence from the 22.235 GHz line. The footprint sizes varied from 148 km by 95 km at 6.6 GHz to 27 km by 18 km at 37 GHz. Values of NE$\Delta$T vary from 0.9 K for the lower frequency channels to 1.5 K at the highest frequency measured.

SMMR shares similarities with equivalent low frequency channels on SSM/I and the patterns are the same, with an increase of all statistics for the 25,000 profile dataset with respect to the training dataset (not shown). Channels 7 and 8 at 21 GHz show the largest differences as they are in close proximity with the 22.235 GHz water vapour line, with a maximum bias of -0.12 K but with a mean value of just above 0.003 K. When all six satellite zenith angles are included mean differences significantly reduce, whereas standard deviations and maximum biases increase for both profile datasets, with a maximum bias of -0.27 K in channels 7 and 8. Brightness temperatures differences are overall low as this is quite a flat part of the spectrum.

### 4.1.7 SSMI/S

The Special Sensor Microwave Imager/Sounder (SSMI/S) is a 24 channel conical scanning instrument that has flown onboard all four DMSP satellites between F16 and F19 (Kunkee et al., 2008). With the most extensive coverage of the microwave instruments validated in this study (and the only one still flying at the time of writing), it is sensitive to a broad variety of atmospheric features and builds on the successes of the previous instruments, as well as including new frequencies for high level sounding. Five channels (12–16) are based on the lower frequency surface sensitive channels of SSM/I between 19.35 – 37 GHz, and five channels (9–11 and 17–18) are based on the higher frequency water vapour channels of SSM/T-2 between 91.66 and 183.31 GHz. The first seven are temperature sounding channels situated in the 50–60 GHz band and five (19–23) are very high resolution channels centred at 60.79 GHz, with a further one (24) at 63.28 GHz, which are sensitive to the upper levels of the atmosphere. The footprint size varies from 74 km by 47 km for the 19 GHz channel to 15 km by 13 km for the 91 GHz channels. NE$\Delta$T varys from 0.2 K for window channels to 1.23 K for channel 24. It should be noted that the Zeeman effect (the splitting of oxygen lines due to Earth's magnetic field), which may influence high peaking channels 19–22 is not modelled in the radiative transfer calculations.

SSMI/S has conical viewing geometry with a fixed zenith angle of 53.1°, so will never utilise the full range of SZA calculated for the standard RTTOV coefficients. To test the accuracy of the validation statistics, which are an average of all six SZA, Fig. 7 shows these values alongside just the fourth SZA, which is equal to a secant of 1.75 and an angle of about 55°, so very similar to SSMI/S. The biases are remarkably similar in most cases, indicating that the average of all SZA biases can be used to accurately represent the true viewing geometry biases, if needed. As many of the channels are the same, or similar, to channels in the previous instruments discussed, there are no big surprises in terms of behaviour. The water vapour channels 7–11 around 183.31 GHz show the biggest differences between RTTOV and AMSUTRAN, and between the training profile set and the



25,000 profile set, as expected based on corresponding channels on SSM/T-2. The higher peaking channels 21–24 around 60 GHz show only slightly bigger differences than the lower temperature sounding channels 1–7 between 50 – 60 GHz. These

follow the same pattern as the MSU temperature sounding channels, however, with lower standard deviations for the 25,000 profile dataset than the training profiles.

### 4.2 Spatial variation of bias from the independent dataset

In forming the statistics shown in the previous section there are details lost in the averaging process that are revealed with a spatial view of the biases over the entire globe, so spatial variability for each profile in the independent dataset was also

evaluated. For the infrared, 2 IRIS-D channels, one SIRS-B and one MRIR channels are shown. The three IRIS-D channels have a corresponding similar channel on another instrument (two in SIRS-B and one in MRIR) to test the robustness of the results. These comprise one surface channel (centred at 899 cm$^{-1}$) one temperature ($CO_2$) channel (centred at 679 cm$^{-1}$) and one water vapour channel (centred at 1510 cm$^{-1}$).

 Figure 8 shows the spatial distribution of the difference between RTTOV and LBLRTM simulations for all independent

profiles. Figures 8a and 8b represent the window channel centred at 899.66 cm$^{-1}$ from IRIS-D and SIRS-B, respectively. The IRIS-D spatial distribution presents a positive bias in the equatorial region and a negative bias in the polar regions. The maximum value reached is 0.05 K (Fig. 8a). The corresponding window channel in the SIRS-B instrument at 899 cm$^{-1}$ has a negative bias (around -0.04 K) dominant in the equatorial regions, but in the rest of the globe it is close to zero (Fig. 8b).

 Figures 9a and 9b show the latitudinal distribution of the channels centred at 679 cm$^{-1}$ and 899 cm$^{-1}$ from the IRIS-D and

SIRS-B, respectively. The figure clearly shows that the bias has latitudinal behaviour, mainly for channel centred at 679 cm$^{-1}$ (black circles). This channel presents a positive bias in all regions and the values are larger in the polar regions and there is an increase of the differences from the extratropical regions to the equatorial region, which is more evident in SIRS-B. The channel centred at 899 cm$^{-1}$ tends to be negative in the equatorial region in the SIRS-B sensor (blue circles), whereas the corresponding channel in IRIS-D has a bias closer to zero, or slightly positive, in the equatorial region. The spatial variability

of SIRS-A channels (not shown) is similar the ones showed for SIRS-B channels. The correlation between the mean bias and the IWVC is moderate for the 899 cm$^{-1}$ channels in the different sensors, however it is positive for the IRIS-D channel (0.48) and negative for SIRS-B (-0.47). There is no correlation (0.017) between mean bias and the IWVC for the channel 679 cm$^{-1}$ of IRIS-D and the correlation is 0.40 for same channel of SIRS-B. The reason for these differences is not entirely clear, but as the only difference between simulations of equivalent channels for both instruments are the bandwidths, with SIRS-B channels

around a factor of 10 wider than IRIS-D, this is likely to be the cause.

 A similar evaluation was made for one other channel from IRIS-D (centred at 1051.10 cm$^{-1}$) and one corresponding channel from MRIR (centred at 1051.03 cm$^{-1}$). Figure 10a and 10b show the spatial distribution of the water vapour channels centred at 1510 cm$^{-1}$ from IRIS-D and MRIR, respectively. Both channels present a positive bias in all regions, but there are a few points with negative bias (for example in the North of Mexico). The values are higher for MRIR and it is possible to see a

difference in intensity between the equator regions and the polar regions. IRIS-D presents positive values in the equatorial regions and predominance of negative and close to zero values in the polar regions. In both cases, larger values are present in



the equatorial region, which is possibly related to the content of water vapour in the atmosphere and its higher variability in these regions. There is no correlation (0.06) between mean bias and the IWVC for the channel 1510 cm$^{-1}$ of IRIS-D and the correlation is weak (0.23) for same channel of MRIR.

For the microwave we examine the spatial distribution of the bias with channels sensitive to a range of features in the microwave spectrum: a water vapour channel centred at the 183.31 GHz line, a window channel at 85.5 GHz and a temperature channel centred at 60.79 GHz. The spatial bias distribution of the water vapour channel on SSM/T-2 with the closest proximity to the 183.31 GHz line (identical to channel 11 on SSMI/S) is shown in Fig. 11a, along with the latitudinal distribution of the bias in Fig. 11b. The bias appears to be strongest in the subtropical belts (particularly the southern one) around 30 °N/S, up

to about 0.5 K. The distribution of the bias around zero K is reasonably symmetrical but there are a few profiles with very negative biases around the equatorial regions, including the maximum bias of nearly -2 K. This is possibly due to the unusual shape of the water vapour profile in the region of deep convective clouds which could be challenging for the RTTOV predictors, but because there is not a correlation between integrated water vapour content and bias (calculated value 0.013) this does not appear to relate to the vertically integrated amount of water vapour in total.

A similar story is seen for the window channel 7 at 85.5 GHz on SSMI shown in Fig. 12, but with a far smaller magnitude. Apart from the surface, window channels will be affected only by the water vapour continuum (in the clear-sky) whose contribution increases smoothly with frequency. The spatial pattern of positive bias is less concentrated in the subtropical belts and more broadly positive overall. In this channel the very negatively biased equatorial profiles have slightly higher integrated water vapour contents than those with differences closer to zero. There is a slight to moderate correlation of 0.45. This pattern

is detected in all microwave window channels with correlations up to 0.49 at 37 GHz. Biases remain at 0.02 K or below for these channels in almost all cases.

For the high peaking temperature channels on SSMI/S the pattern is quite different, see Fig. 13. The biases are mostly negative and less latitudinally variable, apart from some stronger negative values around 60°N/S. There is no correlation with IWVC, a feature that is seen with all microwave temperature sounding channels that sense above the surface. The maximum

difference is -0.18 K at 70°S but the value is anomalously low compared to the rest of the profiles. This is a feature common amongst most channels: the maximum difference is far more extreme than the vast majority of profiles. The profiles responsible for these extreme biases are examined in the following section.

## 4.3    Profiles associated with maximum bias

In order to examine the conditions under which the largest deviations between RTTOV and AMSUTRAN occur, the profiles

associated with each channels maximum bias are shown in Fig. 14–18 for each instrument in turn, and the location of these profiles is shown in Fig. 19. As might be expected some profiles are associated with multiple channels.

For MSU (Fig. 14), profiles for channels 2-4 deviate significantly (lower) from the mean profile (black dashed line), whereas channel 1 is more similar. Profiles associated with channels 2-4 are located on the Antarctic Peninsula and the profile for channel 1 is over Australia, see Fig. 19. As these are all temperature sounding channels the unusually low values of temperature





in the troposphere and stratosphere (left panel) are possibly outside of the range of the values the predictors are designed for, and the low stratospheric ozone values (right panel) likely contribute to this.

For SSM/T-2 (Fig. 15) the largest bias of -1.878 K comes from channel 3, and the water vapour profile associated (middle panel) shows a large anomalous spike of low water vapour in the upper troposphere around 200 hPa. The rest of the profile however is above the mean profile, particularly in the 200-300 hPa region, which may instead be the source of the bias. This
profile is situated in the tropical Indonesian region. Two other profiles responsible for the biases in the four other channels are situated over Ethiopia and the Arabian Peninsula, the latter of which has an particularly dry profile.

The six window channels on SSM/I (Fig. 16) are all similarly affected by the same high humidity profile over Ethiopia, which also has high tropospheric temperatures and stratospheric ozone. Water vapour channel 3 (22.24 +0.12 GHz) has its highest bias associated with a profile that has higher water vapour at a slightly higher altitude, and is situated 5 ° north of the
previous profile. All window/water vapour channels on SMMR (Fig. 17) are similarly greatest affected by the first Ethiopian profile. In total half (25) of all the channels in the MW instruments considered are most affected by this profile.

As the first 17 channels of SSMI/S are the same as channels on the other instruments only the six unique high peaking around 60 GHz are considered here. As can be seen in Fig. 18 the temperature profiles deviate strongly from the mean profile at upper levels between 400 – 0.01 hPa. The profiles associated with channels 1 and 2 have higher than average stratospheric
temperatures and are situated over the Southern Ocean and Arctic, respectively. The remaining four channels have lower than average temperatures and are all situated over the Antarctic Peninsula.

## 5   Conclusions

The main objective of this study was to validate RTTOV coefficients using a large independent profile dataset for the historic infrared instruments: IRIS-D, SIRS-B, MRIR and HRIR and microwave instruments: MSU, SSM/I, SSM/T2, SMMR and
SSMI(S). The top of atmosphere (TOA) radiances are computed at high spectral resolution for a large profile data set (NWPSAF 137 levels models profiles dataset interpolated to coefficient levels) using LBLRTM at nadir and AMSUTRAN at 6 angles view. The LBLRTM TOA radiances convolved with the instruments ISRF were compared against the RTTOV simulations for the infrared, whereas for the microwave a tophat/boxcar function is assumed for the passbands. The statistics of the comparison (mean, standard deviation and maximum) of the large profile dataset were then compared with the statistics of the profiles used
to generate the RTTOV coefficients (83 training profiles dataset). The results for the infrared sensors showed that the statistics for the independent profile dataset (25,000 profiles) are similar to those found when using the 83 training profiles. This confirms that it is acceptable to validate the RTTOV coefficients using the same profiles used to generate the coefficients. Differences between RTTOV and LBLRTM are higher in the water vapour channels, where the differences can reach 0.4 K (up to 0.2 K for the standard deviation) in the independent profiles. In almost all channels evaluated in this work the training profiles
show differences smaller than the 25,000 profiles. The maximum differences were also observed in these channels and the values are higher in the independent profiles (up to 6 K). The latitudinal dependence of the bias is found in the channel centred at 679 cm$^{-1}$ from SIRS-B instrument and the range of the bias is higher for multispectral instrument than for hyperspectral



instrument. A similar behaviour is observed in the channel centred at 1510 cm$^{-1}$ of MRIR. Those noticeable differences between channels on different instruments with similar central wavenumbers show the importance of the specification of the 380 ISRF.

For the microwave sensors examined the biggest differences between RTTOV and AMSUTRAN are in water vapour channels, with means of up to 0.02 K (up to 0.1 K standard deviation) in the training profile set, however the validation with 25,000 profiles shows this increases to 0.04 K (up to 0.13 K standard deviation) which is still very low overall. Maximum differences in the training profile set reach -0.3 K in these channels, whereas a value of nearly -2 K was seen in the larger profile set, 385 however these very low values are extremely rare, and are associated with profiles that significantly deviate from the profile mean and located in regions with unique atmospheric conditions, such as deserts, tropics or the polar regions.

Even though this study is restricted to historic sensors, the majority of which are no longer in operation, it confirms that the validation statistics for the 83 profile dataset are adequate to represent the overall biases for a range of different instruments. Equivalent statistics for all sensors supported by RTTOV can be found on the NWPSAF website[4] and can be used to provide 390 an average bias correction. This study has further shown examples of the potential for exploiting error predictors such as satellite zenith angle, ice water content and spatial distribution of the differences, which may help with the development of the bias correction procedure applied to fast satellite simulations, by identifying regions and scenes that challenge RTTOV in reproducing the line-by-line results. The next phase of the C3S project examines the covariances of these biases between channels.

*Code and data availability.* The RTTOV model can be downloaded from the NWP SAF website (https://nwp-saf.eumetsat.int/site/software/rttov/). The LBLRTM model can be downloaded from https://github.com/AER-RC/LBLRTM. The atmosphere profiles can be downloaded from https://nwp-saf.eumetsat.int/site/software/atmospheric-profile-data/.

*Author contributions.* BBS, ECT and JV wrote the document. The simulations were performed by BBS and ECT. BBS and ECT generated the figures.

*Competing interests.* The authors declare that they have no conflict of interest.

*Acknowledgements.* We acknowledge SPASCIA, who led the consortium for the C3S 311c data rescue study that initiated this work, which comprised the University of Reading, CNRM, the UK Met Office and ICARE/AERIS.

---

[4]https://nwp-saf.eumetsat.int/site/software/rttov/download/coefficients/comparison-with-lbl-simulations/



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



| Name | Platform | Channels | Temporal Coverage |
|---|---|---|---|
| Infrared intruments | | | |
| IRIS-D | Nimbus 4 | 400-1600 cm$^{-1}$ | 1970–1971 |
| MRIR | Nimbus 2 and 3 | 4 | 1966–1970 |
| THIR | Nimbus 4 and 7 | 2 | 1970–1985 |
| SIRS | Nimbus 3 and 4 | 8 (+6) | 1969–1971 |
| HRIR | Nimbus 1 to 3 | 1 | 1964–1970 |
| Microwave instruments | | | |
| SMMR | Nimbus 7 | 10 | 1978–1987 |
| SSM/T-2 | DMSP F11 - F15 | 5 | 1992–2015 |
| MSU | TIROS-N - NOAA-14 | 4 | 1979–2007 |
| SSM/I | DMSP F8 - F15 | 7 | 1987–2020 |
| SSMI/S | DMSP F16 - F19 | 24 | 2004–2016 |

**Table 1.** List of instruments studied in the C3S project and evaluated in the present study. Adapted from GSICS newsletter (Vidot et al., 2021)



**Figure 1.** a) Temperature profiles (K) of each 25,000 independent dataset b) temperature profile of training dataset, c) water vapour (ppmv) of idenpendent dataset, d) water vapour of training dataset, e) ozone (O₃) of independent dataset and f) ozone form training dataset in ppmv. Green line represents the mean profile of 25,000. Red, yellow and blue lines represent the mean profile, minimum and maximum profiles of training profiles, respectvely

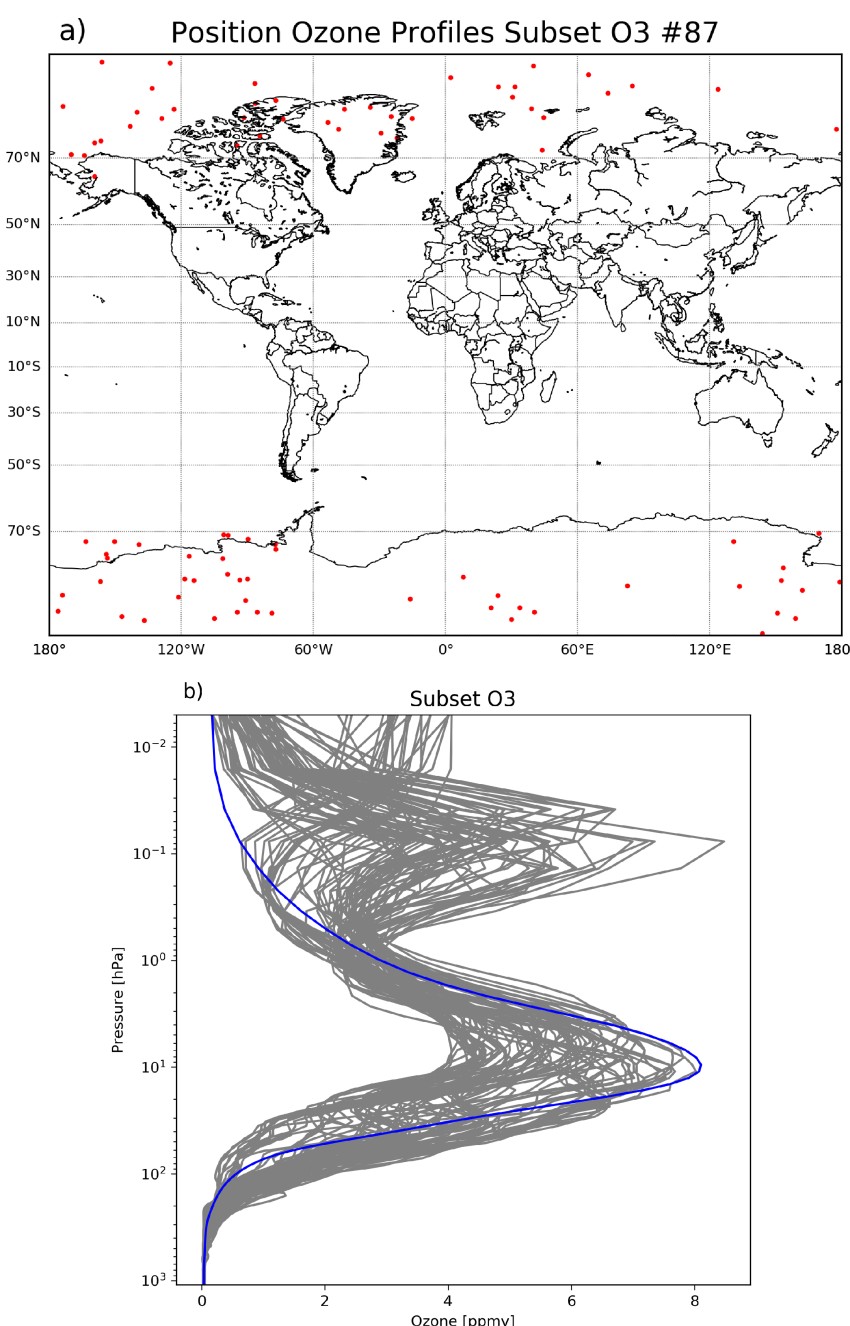

**Figure 2.** The 87 ozone profiles with 3 ppmv maximum above 0.9hPa. a) Spatial distribution of these profiles and b) vertical distribution, the blue line represents the mean profiles of ozone from ozone subset.



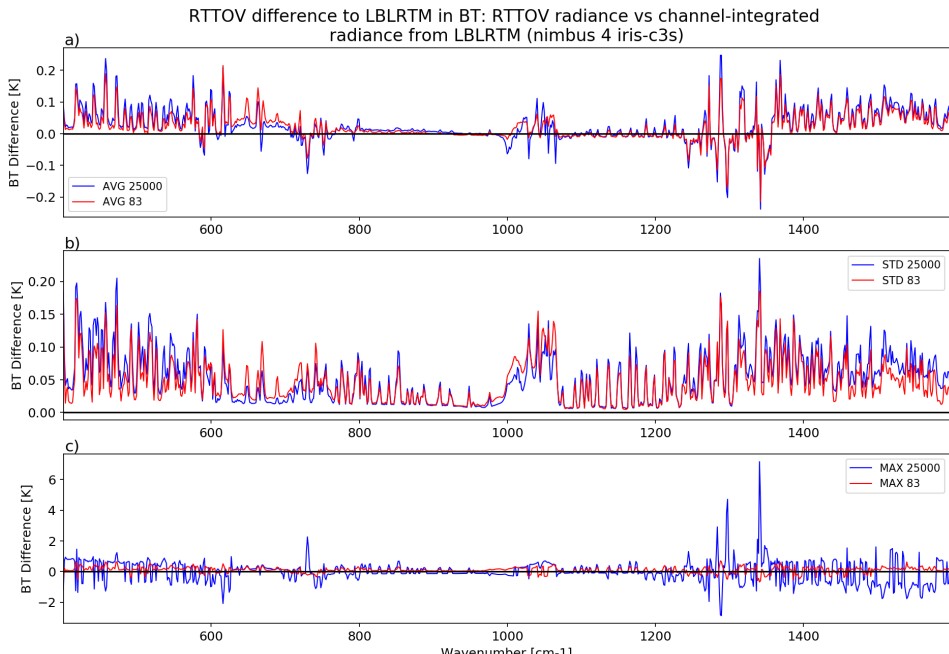

**Figure 3.** Difference between RTTOV and LBLRTM in brightness temperature (K) for IRIS-D in terms of: a) the mean differences, b) the standard deviation of differences and c) the maximum differences. Blue lines represent the independent profiles and red lines are the training profiles.



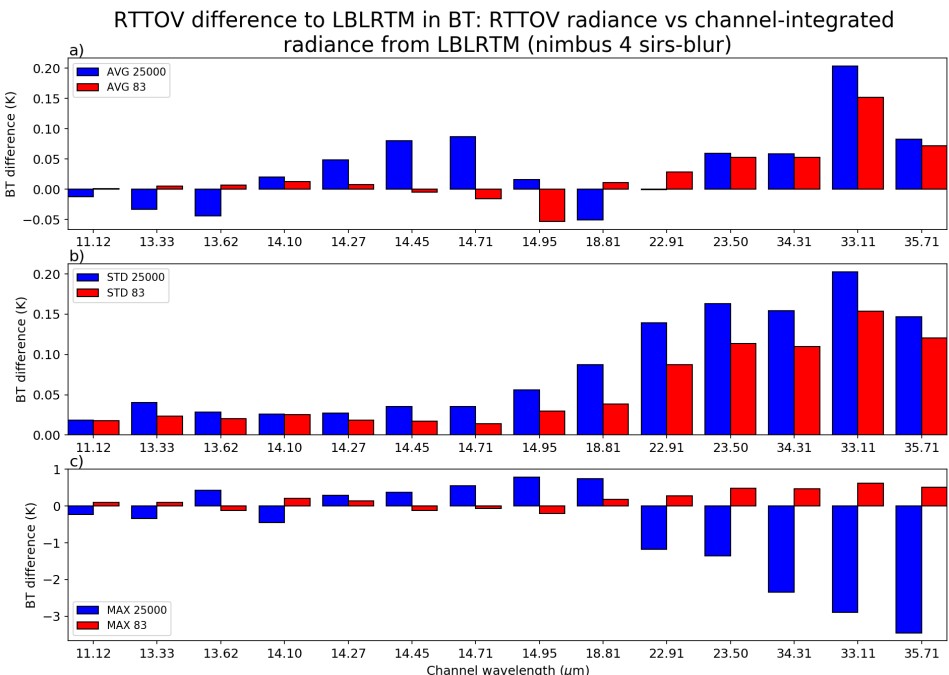

**Figure 4.** Same as Fig.3 for SIRS-B. The channels are represented by bars.



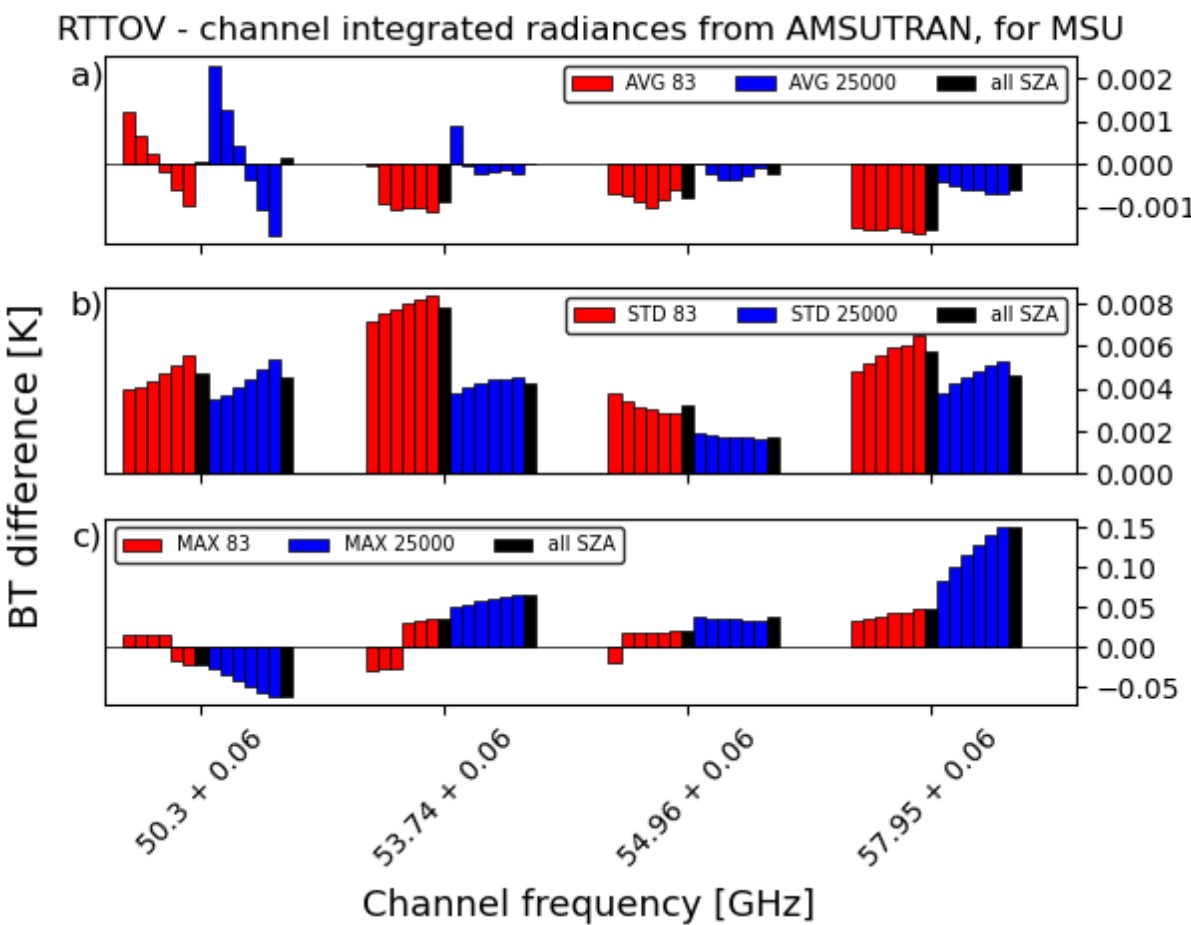

**Figure 5.** Difference between RTTOV and AMSUTRAN for MSU in brightness temperatures [K] in terms of: a) the mean of all profiles, b) the standard deviation and, c) the maximum difference. Blue bars represent the 25,000 profiles and red bars are the training profiles. Each statistic is split by SZA, with nadir on the left, then the five subsequent SZA values up to 63 degrees. The total statistics for all six SZA are then shown in the following black bar, for each of the profile datasets.



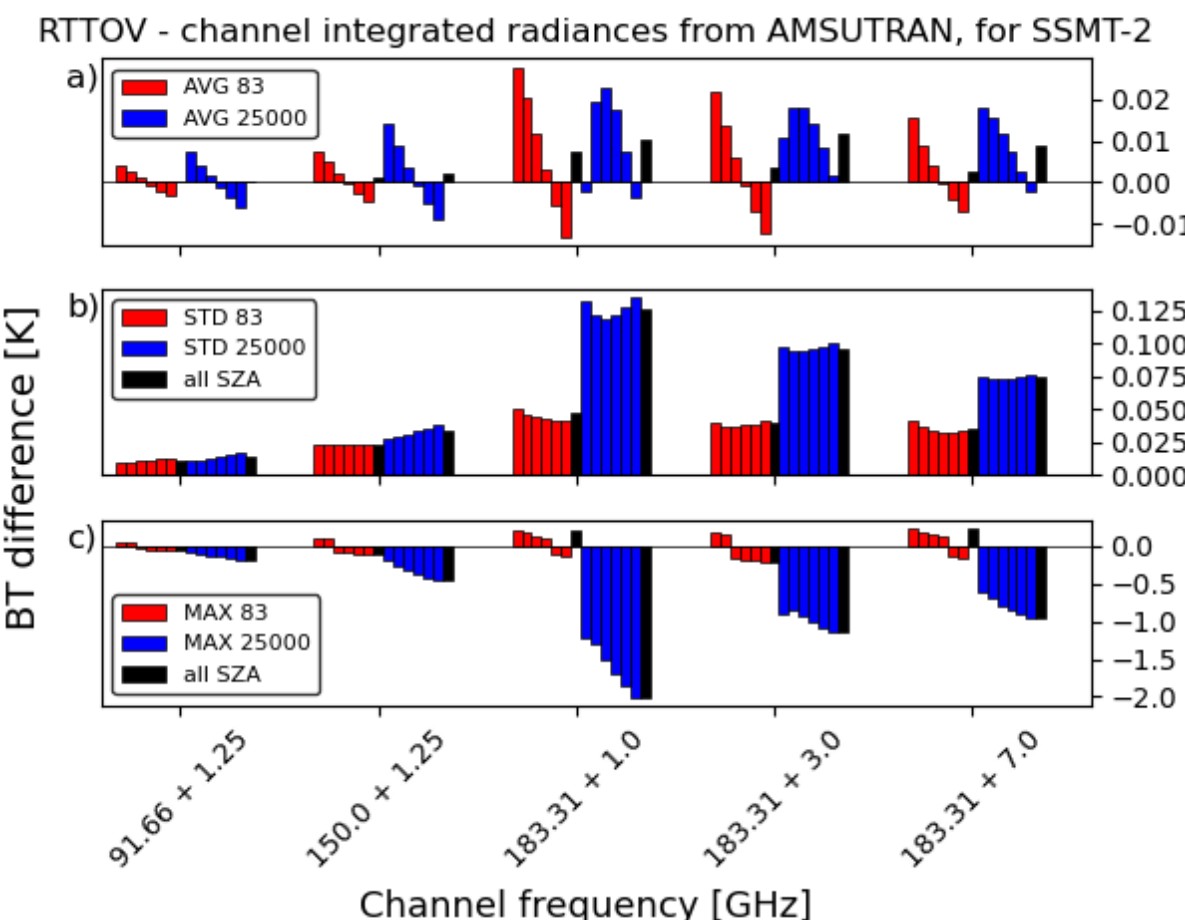

**Figure 6.** Same as Fig.5 but for SSM/T-2.



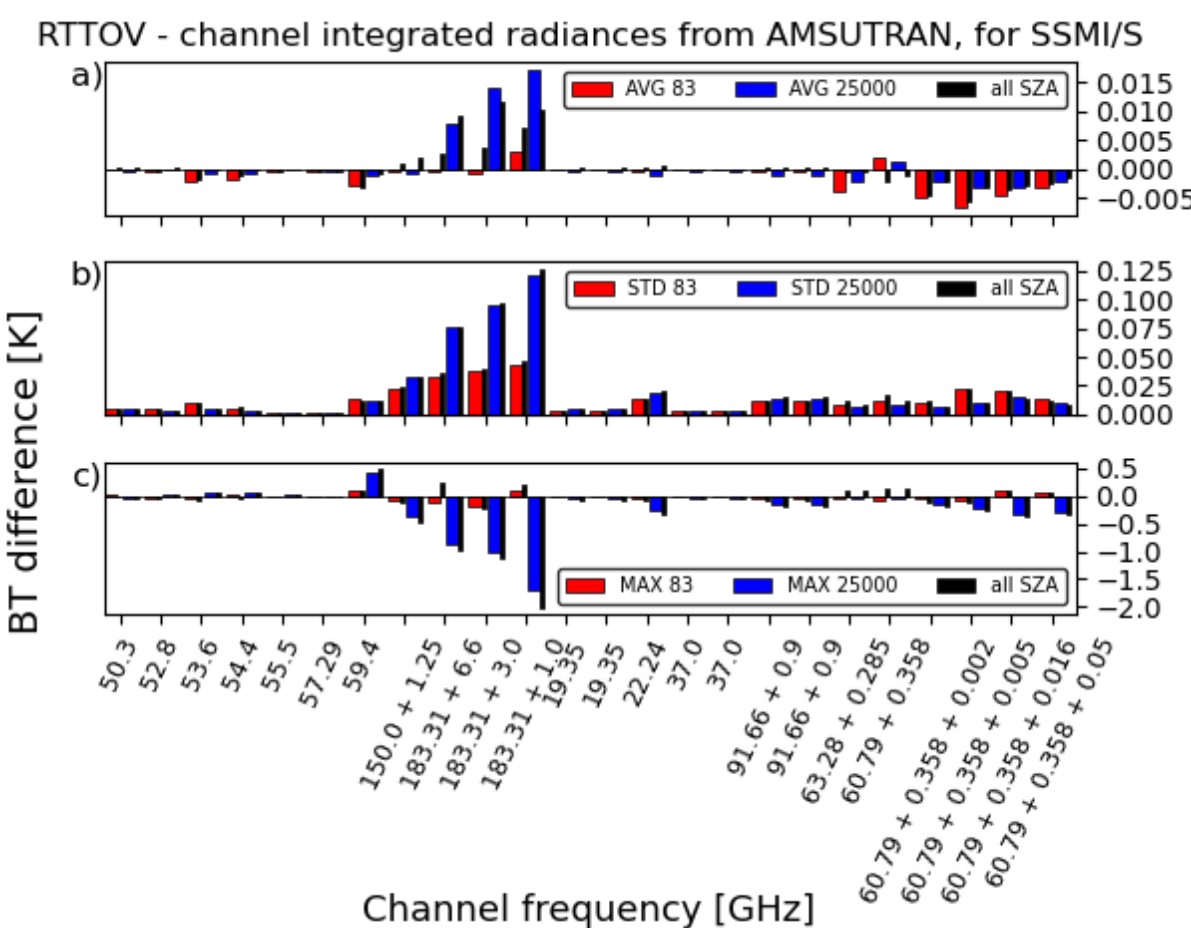

**Figure 7.** As Fig.6 but for SSMI/S. Only the fourth SZA, SECANT=1.75, is shown in the red/blue bars.



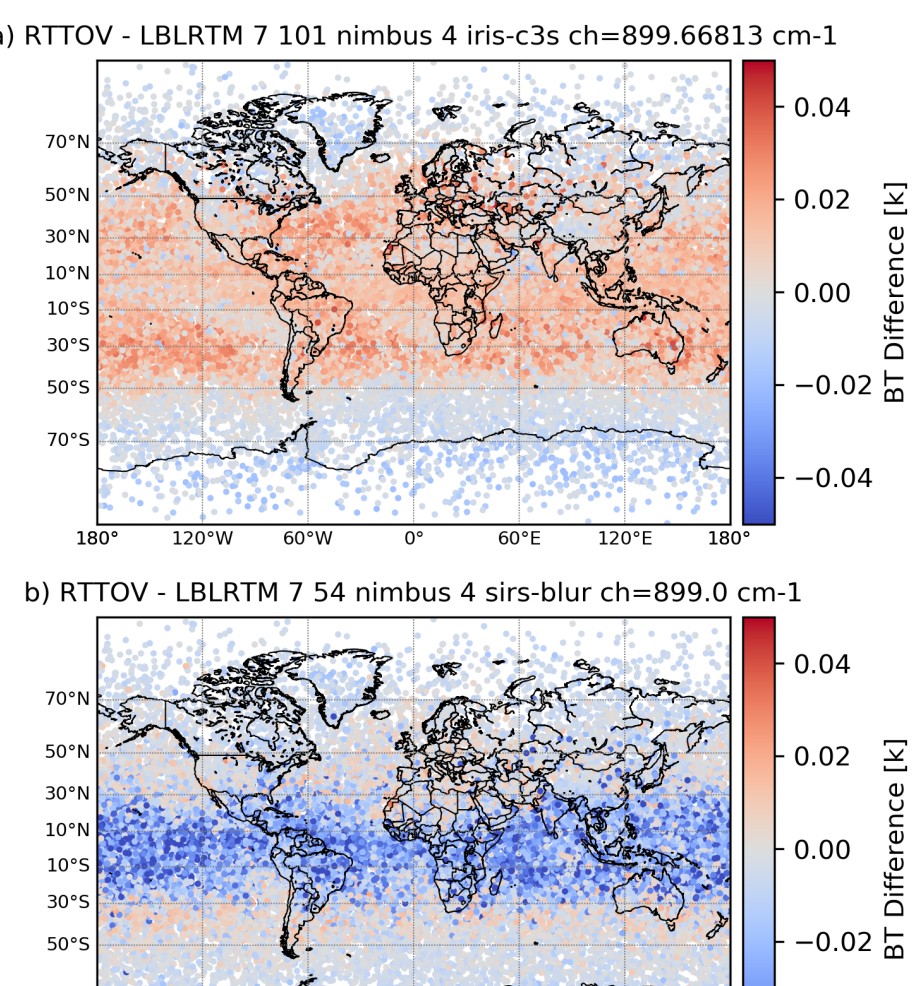

**Figure 8.** Spatial distribution of RTTOV minus LBLRTM for a) channel 899.66 cm$^{-1}$ from IRIS-D and b) channel 899.0 cm$^{-1}$ from SIRS-B. Each point represents one of the 25000 profiles.



**Figure 9.** Latitudinal distribution of the difference between RTTOV and LBLRTM for each independent profile. a) Channels centred at 679.96 cm$^{-1}$ (black circles) and centred at 899.66 cm$^{-1}$ (blue circles) from IRIS-D, b) channels centred at 679.8 cm$^{-1}$ (black circles) and centred at 899 cm$^{-1}$ (blue circles) from SIRS-B. Each point represents one of the 25000 profiles.

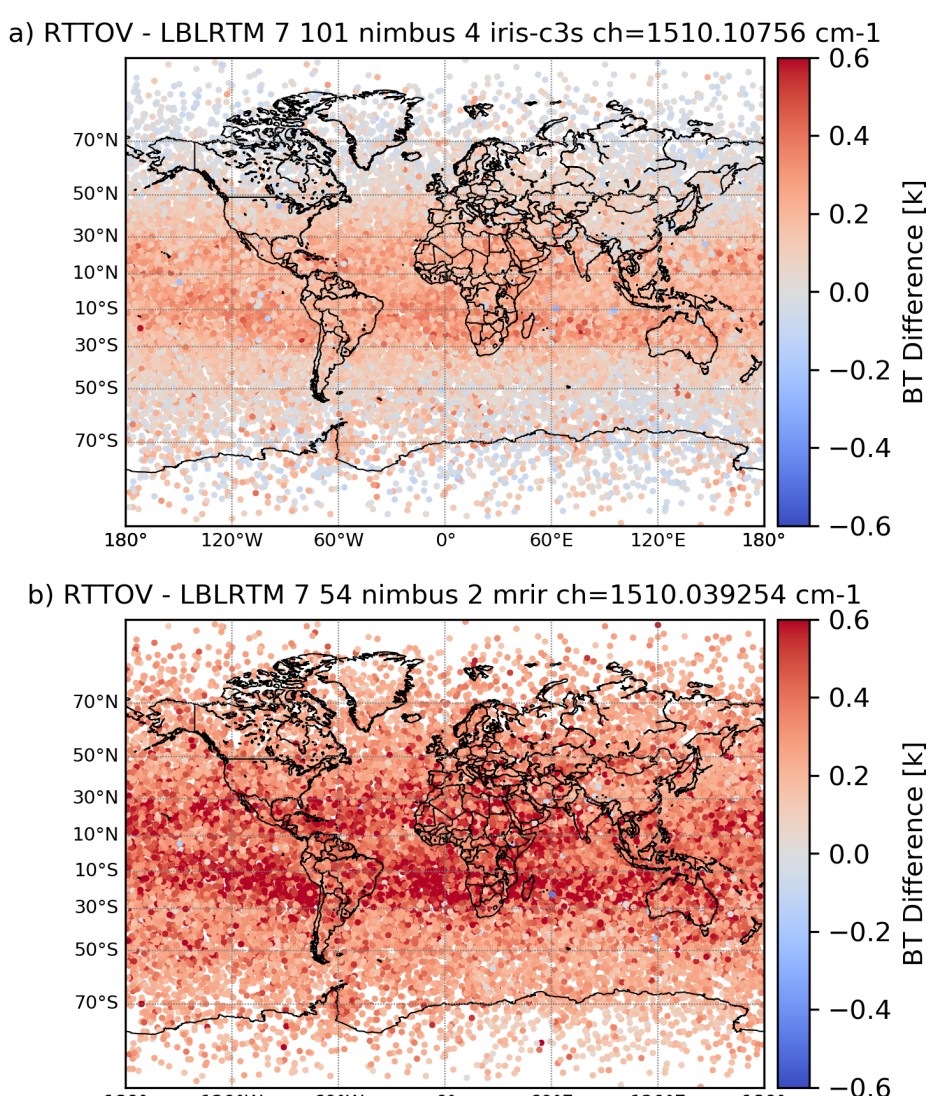

**Figure 10.** Spatial distribution of RTTOV minus LBLRTM for a) channel centred at 1510.11 cm$^{-1}$ from IRIS-D and b) channel centred at 1510.04 cm$^{-1}$ from MRIR. The profiles are from all five subsets.





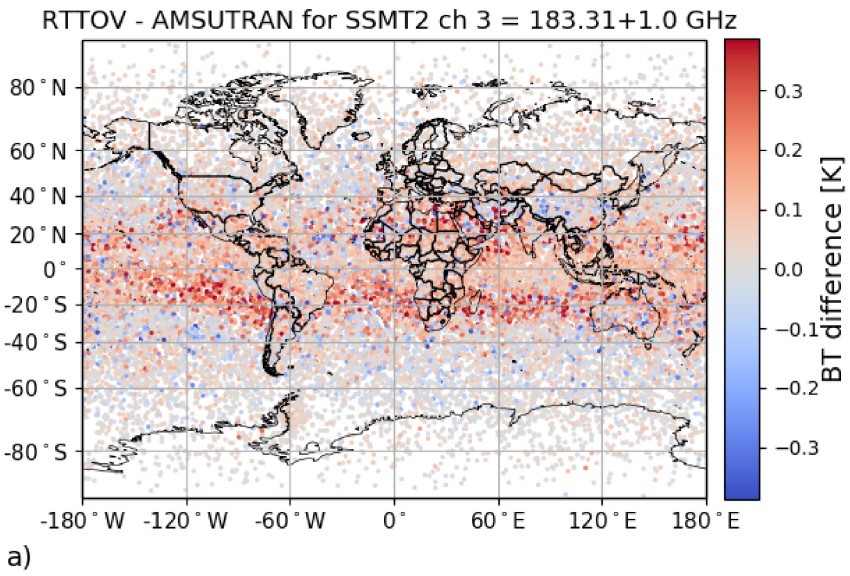

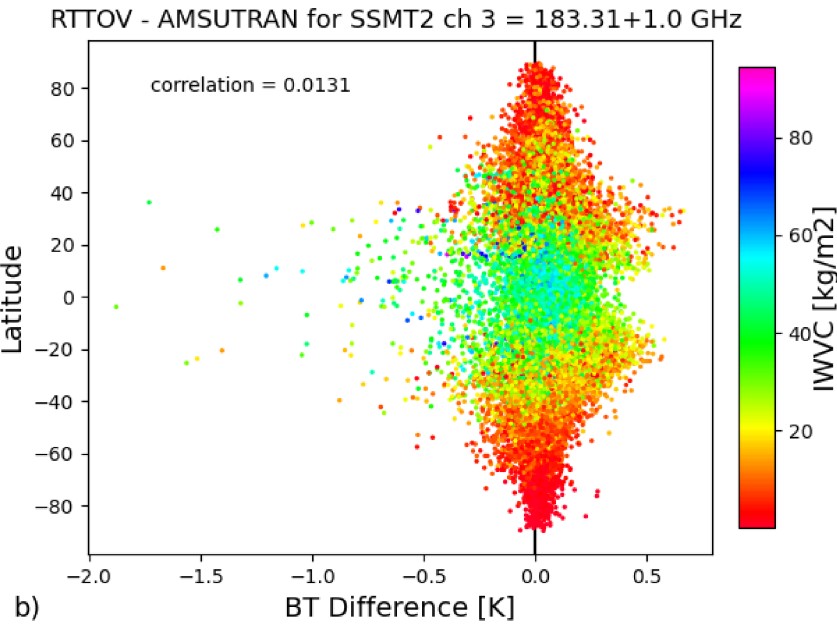

**Figure 11.** a) Spatial distribution of RTTOV minus AMSUTRAN for channel 3 of SSM/T-2 at 183.31 + 1.0 GHz, b) latitudinal distribution of the difference between RTTOV and AMSUTRAN for each profile. The colour bar in b) represents the Integrated Water Vapour Content (IWVC) from all five subsets.





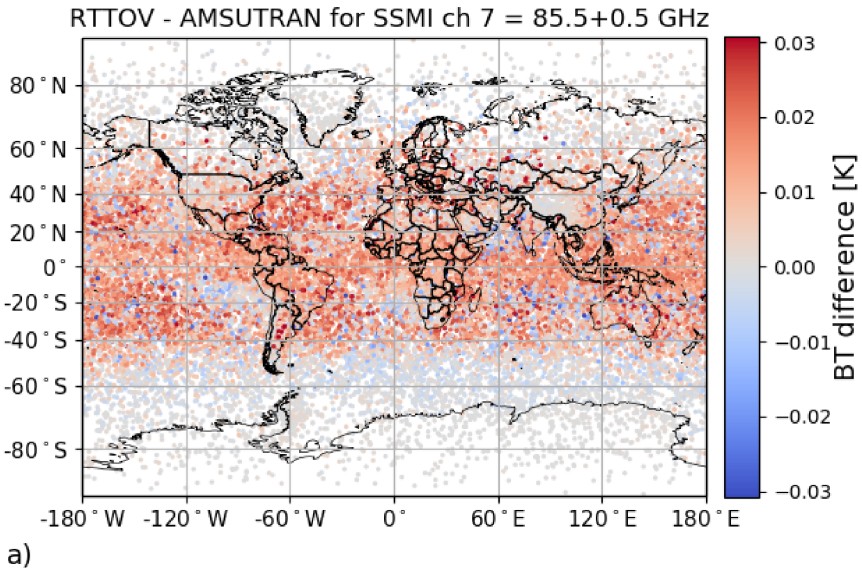

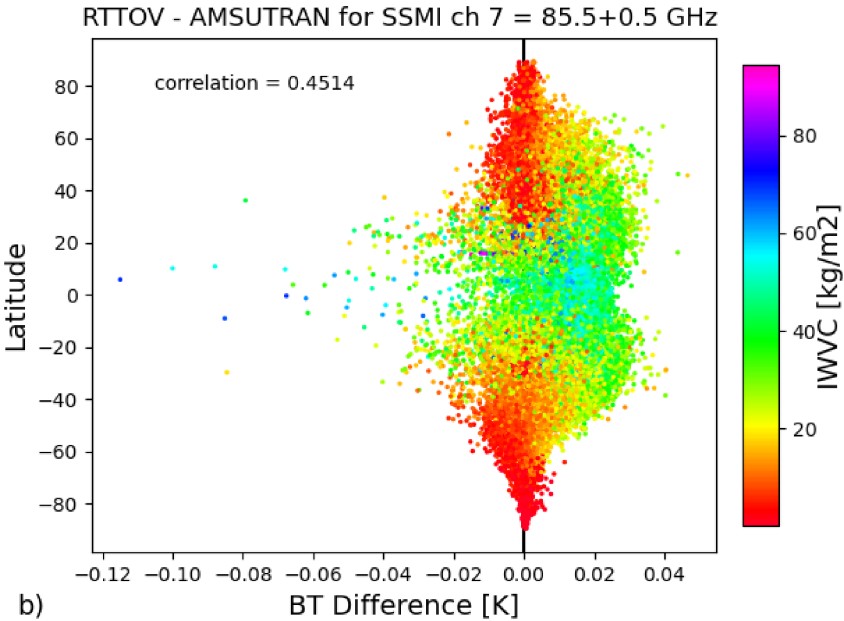

**Figure 12.** a) Spatial distribution of RTTOV minus AMSUTRAN for channel 7 of SSM/I at 85.5 + 0.5 GHz, b) latitudinal distribution of the difference between RTTOV and AMSUTRAN for each profile. The colour bar in b) represents the Integrated Water Vapour Content (IWVC) from all five subsets.





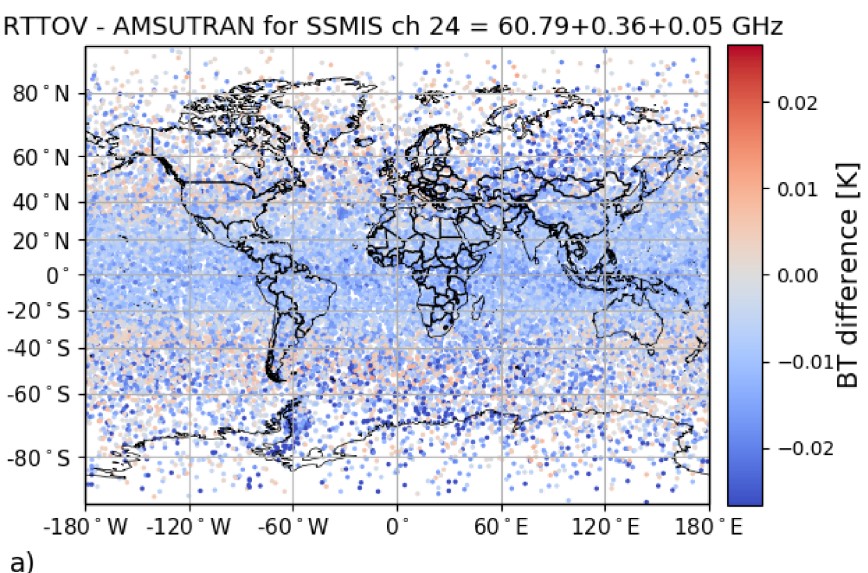

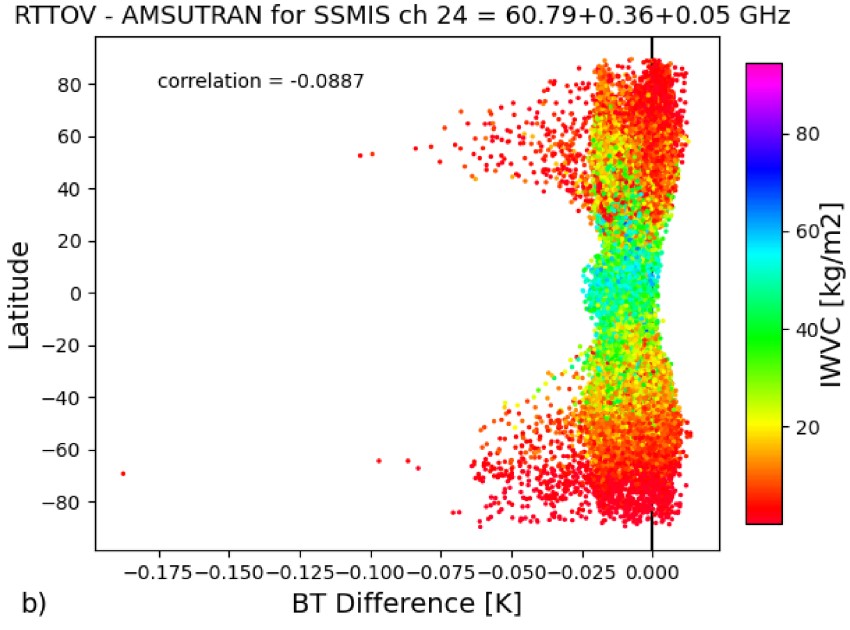

**Figure 13.** a) Spatial distribution of RTTOV minus AMSUTRAN for channel 24 of SSMI/S at 60.79 + 0.36 + 0.05 GHz, b) latitudinal distribution of the difference between RTTOV and AMSUTRAN for each profile.



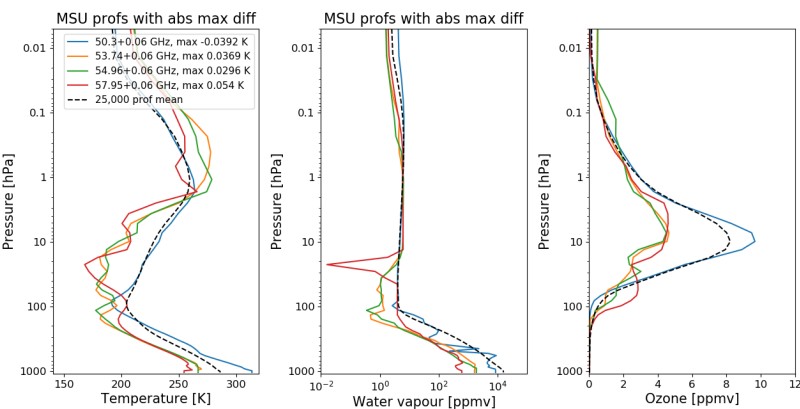

**Figure 14.** Temperature (left), water vapour (middle), and ozone (right) profiles responsible for the maximum bias in each channel of MSU. The dashed black line is the mean of the 25,000 profiles.





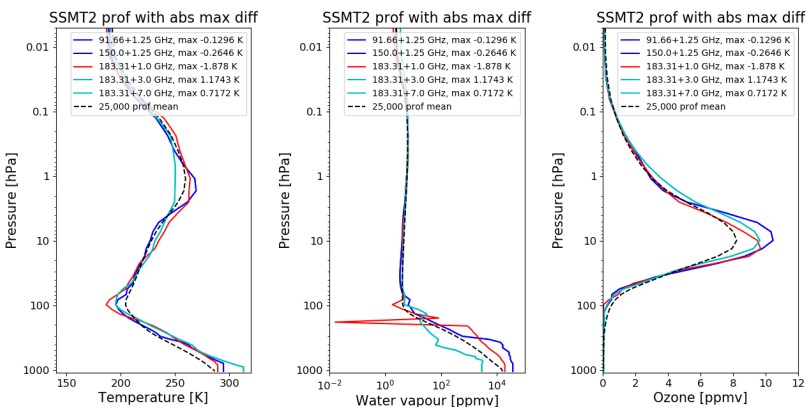

**Figure 15.** Temperature (left), water vapour (middle), and ozone (right) profiles responsible for the maximum bias in each channel of SSM/T-2. The dashed black line is the mean of the 25,000 profiles.





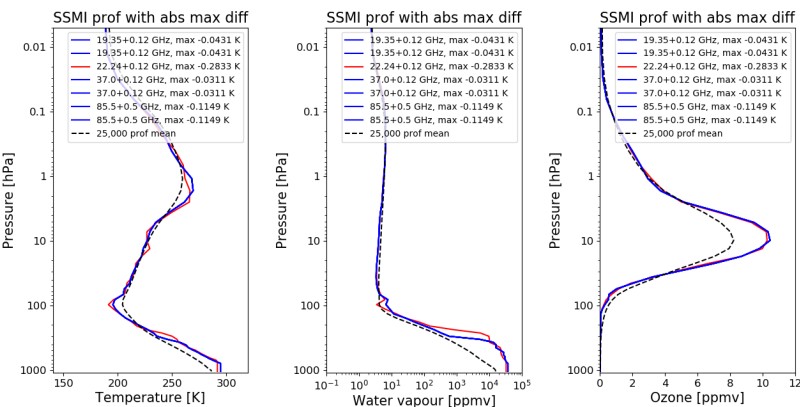

**Figure 16.** Temperature (left), water vapour (middle), and ozone (right) profiles responsible for the maximum bias in each channel of SSM/I. The dashed black line is the mean of the 25,000 profiles.



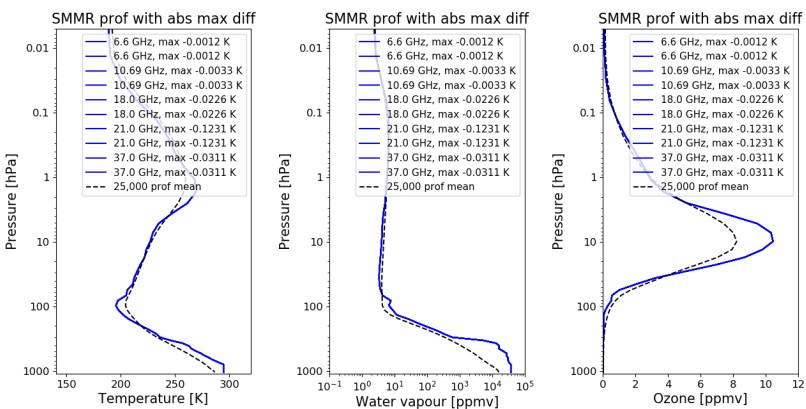

**Figure 17.** Temperature (left), water vapour (middle), and ozone (right) profiles responsible for the maximum bias in each channel of SMMR. The dashed black line is the mean of the 25,000 profiles.





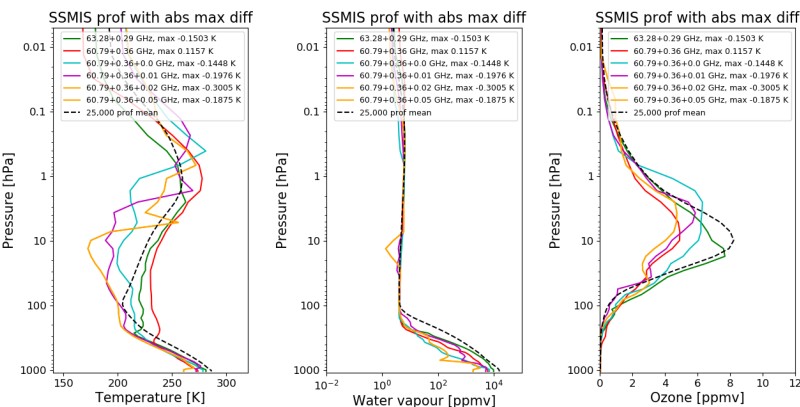

**Figure 18.** Temperature (left), water vapour (middle), and ozone (right) profiles responsible for the maximum bias channels 18-24 of SSMI/S (high peaking temperature channels). The dashed black line is the mean of the 25,000 profiles.

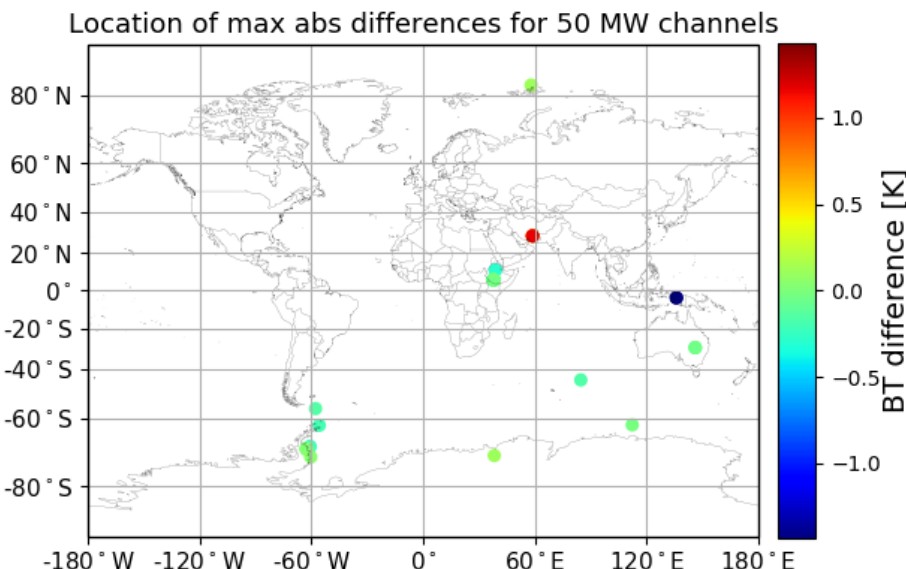

**Figure 19.** Global location of profiles responsible for each of the maximum biases for the 50 combined channels of the five MW instruments in this study. There are less than 50 points are some are associated with multiple channels.