# Peer review of "Global evaluation of RTTOV coefficients for early satellites sensors"

_Atmospheric Measurement Techniques, 2023_

## Author Response (AR1)

**Reviewer 1:**

RTTOV is a fast radiative transfer (RT) model used among many other things for data assimilation in many NWP centers. As such, it requires to be fast but accurate. To calculate gas absorption, RTTOV relies on linear regression methods to use pre-trained absorption coefficients for each instrument of interest in combination with a set of predictors. A way to evaluate the error of this simplification is to compare RTTOV simulations with more computationally demanding models that solve absorption on the fly (line-by-line - LBL - models). The NWP Satellite Application Facility (NW-PSAF) routinely makes this comparison using a set of 83 profiles - the same profiles that are used to generate the instrument absorption coefficient files. The authors argue that the robustness of such validation should be tested by using a larger set of profiles. Hence, the work presented compares RT simulations for these 83 profiles and for 25,000 globally distributed profiles, and argues that it is accurate enough to use the smaller set of profiles. The submitted publication evaluates nine MW and IR historic satellite instruments: IRIS-D, SIRS-B, MRIR, HRIR, MSU, SSM/I, SSM/T2, SMMR and SSMI/S - still relevant today in terms of their frequencies in the observation system. The evaluation is very much relevant for publication after the following comments are addressed.

**We thank Reviewer 1 for his/her comments, questions and remarks, which we hope helped to improve the quality of the original manuscript. Please find below our responses to your comments. Reviewer 1's comments are in** normal **font, our answers are written with bold font.**

Introduction

L9: ISRF. First time you mention it. Please use "Instrument Spectral Response Functions (ISRF)"

**L9: We agree with you. We changed the text by "Examination of the latitudinal dependence of the bias reveals different patterns of variability between similar channels on different instruments, such as 679 cm-1 on both IRIS-D and SIRS-B, showing the importance of the specification of the Instrumental Spectral Response Functions (ISRF)."**

L12: Too many parenthesis " (Radiative Transfer code for TOVs (TIROS Operational Vertical sounder))"

**L13: We agree with you. We changed in the text. "The fast radiative transfer model RTTOV - Radiative Transfer code for TOVs (TIROS Operational Vertical sounder), Saunders et al., (2018), is used as the observational operator that assimilates satellite measurements in multiple Numerical Weather Prediction (NWP) models (i.e., Eyre et al., 2022), enables the retrieval of atmospheric or surface parameters (i.e., Merchant et al., 2019) or the simulation of satellite imagery from NWP models,**

**and is also widely used across the world as a stand alone model for scientific research applications (i.e., Chen and Bennartz,2020)."**

L14: replace by "(i.e., Eyre et al., 2022)"

**L15: The suggestion were applied as follows. "The fast radiative transfer model RTTOV - Radiative Transfer code for TOVs (TIROS Operational Vertical sounder), Saunders et al., (2018), is used as the observational operator that assimilates satellite measurements in multiple Numerical Weather Prediction (NWP) models (i.e., Eyre et al., 2022), enables the retrieval of atmospheric or surface parameters (i.e., Merchant et al., 2019) or the simulation of satellite imagery from NWP models, and is also widely used across the world as a stand alone model for scientific research applications (i.e., Chen and Bennartz,2020)."**

L16: replace by "(i.e., Chen and Bennartz, 2020)"

**L17: The suggestion were applied as follows. "The fast radiative transfer model RTTOV - Radiative Transfer code for TOVs (TIROS Operational Vertical sounder), Saunders et al., (2018), is used as the observational operator that assimilates satellite measurements in multiple Numerical Weather Prediction (NWP) models (i.e., Eyre et al., 2022), enables the retrieval of atmospheric or surface parameters (i.e., Merchant et al., 2019) or the simulation of satellite imagery from NWP models, and is also widely used across the world as a stand alone model for scientific research applications (i.e., Chen and Bennartz,2020)."**

L17: replace combines by combine

**L19: The suggestion were applied. "The `fast' nature of RTTOV is attributed to the linear regression methods at its core, which combine pre-trained satellite coefficients with various combinations of predictors for each atmospheric constituent, in place of the full line-by-line atmospheric absorption calculation."**

L17: please specify that the satellite coefficients are satellite gas absorption coefficients

**L18: The suggestion were applied as follows. "The `fast' nature of RTTOV is attributed to the linear regression methods at its core, which combine pre-trained satellite gas absorption coefficients with various combinations of predictors for each atmospheric constituent, in place of the full line-by-line atmospheric absorption calculation."**

L19: please rephrase. Perhaps it would be more appropriate to cite Saunders et al., 2007 first as work that has analyzed the accuracy of the RTTOV approximations. Please also give an order of magnitude of the "small errors". The paper submitted also contributes to the evaluation of such accuracy. Please also justify the need for further analysis of such accuracy.

**L20: We rephrase the sentence by "The accuracy of the RTTOV transmittance parameterisation was firstly analysed by Saunders et al. (2007) that shown an overall agreement within 0.05 K between different RTM and LBL models, except for certain spectral regions. The evaluation was based on a subset of 49 atmospheric profiles selected from a large atmospheric profiles dataset of the ECMWF ERA-40 database. This first study reported the spectral consistency of the LBL and RTM models but the few numbers of profiles used did not provide insights on the global distribution of the difference."**

L34: The sentence starting with "Current validation […]" needs rephrasing. It could be actually discussed together with the Saunders et al., 2007 paper as the aims are similar.

**L25: We rephrase the sentence and change the place in the text by "Current validation of RTTOV coefficients of clear-sky simulations is based on the comparison between LBL simulations versus the same results from RTTOV. The standard 83 training profiles used for coefficient generation are used in this validation."**

L35: Please rephrase "and statistical plot and data"

**L40: We rephrase the sentence by "This validation is done for all instruments simulated by RTTOV and the associated statistical data and figures can be found on the NWP Satellite Application Facility (NWPSAF) website."**

L48: Please rephrase "The satellite instruments […]"

**L54: We rephrase the sentence by "Validation results, in terms of the mean, standard deviation and maximum differences between LBL and RTTOV simulations for both the 25,000 independent profiles and the training profiles will be presented in Section 4 for each satellite instrument."**

Atmospheric profiles

L42: Too many parentheses here too.

**L45: We do not see too many parentheses as two parentheses are needed for the acronym of each instrument.**

L56: Replace instruments for instrument.

**L63: We replaced it. "The diverse profile training dataset contains 83 profiles for six molecules (water vapour ($H_2O$), ozone ($O_3$), carbon dioxide ($CO_2$), methane ($CH_4$), nitrous oxide ($N_2O$) and carbon oxide (CO)) and one standard profile, mostly from the US76 standard atmosphere database (1976), is used for the other 22 molecules and Chlorofluorocarbons (CFCs), though not every molecule is included for every instrument, depending on its spectral absorption coverage."**

L57: Were these 80 profiles selected randomly? In terms of location and time of the year? What is the top of the model? The same question for the 25,000 profiles selected from the IFS.

**L70: The 80 profiles were not selected randomly but following a strategy to represent as best as possible the range of variation of temperature and absorber amount of the real atmosphere using the ECMWF IFS model Cycle 30R2 during the period July 2006-July 2007. The top of the model is 0.01 hPa both for the 83 profiles and for the 25,000 profiles dataset. We added the phrase: "The profile is split into 137 levels from the surface to 0.01 hPa (training profiles present the same top of model), which is the resolution currently used by the Integrated Forecasting System (IFS) developed at ECMWF."**

L67: What do you mean by "Each of the 5,000 profile subsets represents maximum diversity of one of five different variables […]"?

**L73: We mean that each subset of profiles displays maximum variability of one of the five different (temperature, humidity, ozone, clod condensate and precipitation) variables. We change the paragraph. "The 5 subsets of 5,000 profile represent maximum variability of one of five different variables: temperature (t), specific humidity (q), ozone (o3), cloud condensate (ccol) or precipitation (rcol)."**

L83: "The small values of ozone in the training profiles could be due to a profile located at the ozone hole". In connection with the question above about L57: are training profiles all from the same area?

**The 83 training dataset are from global dataset, as explained above. These profiles were constructed to be representative of a variation of atmosphere conditions, as explained by Matricardi(2008).**

Line-by-line models and RTTOV

L91: "RTTOV setup" title. Perhaps it is more appropriate to use simply "RTTOV radiative transfer simulations" ?

**L199: We agree with your suggestion. And we also changed the order of the sub-sections 3.1 and 3.2. A new section was added, so the sub-sections 3.1 and 3.2 are now 4.1 and 4.2. .**

L92: What do you mean by profiles failing the LBL models? Perhaps it would make more sense also to introduce line by line models first?

**L161: These were profiles that fell outside of some internal checks that LBLRTM imposed on profile limits. Using a more recent version of LBLRTM revealed that these checks had been modified and the same profiles no longer failed. As it was only a few profiles out of 25,000 their omission will likely have had no effect on the**

**resulting statistics, so we suggest removing this comment to avoid confusion. We have rearranged the sections to introduce line-by-line models first (sub-section 4.1).**

L94: Here it gets a bit confusing regarding instruments and pressure levels.

**L202: "For hyperspectral instruments 101 levels is required as the full vertical stratification of the atmosphere is resolved, but 54 levels is sufficient for all other narrowband instruments (SIRS, MRIR, HRIR, MSU, SSM/I, SSM/T-2, SMMR, SSMI/S)."**

Some comments:

• L95: you have not defined which instruments are narrowband yet.

**L202: We change the sentence to ""For hyperspectral instruments 101 levels is required as the full vertical stratification of the atmosphere is resolved, but 54 levels is sufficient for all other narrowband instruments (SIRS, MRIR, HRIR, MSU, SSM/I, SSM/T-2, SMMR, SSMI/S)."**

• It is not clear as to Why you have to use different profile levels for different instruments?

**L202: Changed a word in - "For hyperspectral instruments 101 levels is required as the full vertical stratification of the atmosphere is resolved, but 54 levels is sufficient for all other narrowband instruments (SIRS, MRIR, HRIR, MSU, SSM/I, SSM/T-2, SMMR, SSMI/S)."**

• Also no reference previously in the text to different versions of RTTOV predictors.

**L204: We changed in the text as follows: "For the infrared instruments the version 7 predictor RTTOV coefficients (with 101 levels for IRIS-D and 54 levels for other instruments) were used because with these there is no variation in the carbon dioxide ($CO_2$) profile. Version 7 predictors were introduced at the release of RTTOV-7, and are a development of those used in RTIASI. There are 10 predictors specified for the mixed gases (e.g. $O_2$ and $N_2$), 15 for water va pour and 11 for ozone, and they are functions of satellite zenith angle, temperature, water vapour and ozone mixing ratio, see Table 1 in Saunders et al. (2002) for full details and Saunders et al (2018)."**

L97: "As with the LBLRTM simulations". These have not been discussed yet. Perhaps it makes more sense to introduce these before the RTTOV setup.

**We changed the order of section 3.1 and 3.2. Now "Line-by-line radiative transfer model setup" is section 4.1 and "RTTOV radiative transfer simulations" is section 4.2.**

L103: Why are 54 levels sufficiently accurate for this analysis?

**L214: We have added the following explanation to the text: "All microwave RTTOV simulations use version 7 predictors, and 54 (as opposed to 101) levels are**

**sufficiently accurate for these instruments, as demonstrated by Saunders et al. (2013)."**

I have some additional questions regarding this section:

• You are including CO2 in the simulations. In the independent profiles the CO2profile is always the same (CO2 in profile 83), but in the training profiles 1-82 CO2 varies. Is this correct? Why not exclude CO2 directly?

**L97: The CO2 profiles are varying in the training profiles, but we can either fix it or not in the training for the coefficients generation and then for the simulations, depends on what we want. If we want the CO2 profiles to vary, because we have CO2 profiles information as input of the simulations, then we use coefficients that were trained with varying CO2. If not, we fix the CO2 (profile 83) for the training. In this study, as we compare the simulations with 25,000 profiles without varying CO2, we used coefficients that were trained with fix CO2. To clarify this, we added the following sentence at the end of the part 2 on atmospheric profiles (L97): "It is worth mentioning here that the RTTOV coefficients generated for the study were the ones with only water vapour and ozone as variable gases in order to be used for the comparison with the 25,000 diverse atmospheric profiles dataset". The effect of other trace gases were not evaluated in this study such as from CO2, CH4, N2O, CFCs, etc... Even the concentration of these gases have increase in the atmosphere since the 1970s, the impact of them in RTTOV simulations is behind the scope of this study. However, for reanalyse applications the RTTOV coefficients for IR instruments were generated with variable CO2 (Poli et al., 2017) in order to take into account its increase during the last decades.**

• What about the other minor gases that are in the training dataset: CH4, N2O, CO? You mention something in the IR LBL RT model, but not here.

**We used a mean profile of those minor gases in our simulations both for LBL or RTTOV models. They are considered fixed for all simulations.**

L105: It would help the reader to mention in the introduction that different radiative transfer models were used for the IR and the MWs.

**L40: We added the paragraph: "Different LBL models are used for infrared and microwave sensors".**

L128: Where do you get Nitrogen and ozone from? the mean training profiles? Please specify.;

**L168: These information are mentioned in paragraph "As previously mentioned, the independent profiles only contain temperature, water vapour and ozone information, and profiles relating to the molecules: CO2, CH4, N2O and CO are the**

**mean training profile set (profile 83), and one US76 standard profile for other the 22 other molecules.**

Independent profile dataset versus training profile dataset

L145: "A description of each of the sensors used in this study follows". In this section you describe the sensors. All this information should be in another section, rather than in the same section of the results. For example L151, L167, L198-201, etc.

**L99: We created a "Satellite sensors" section which presents a brief description of IR and MW sensors.**

L148: "Note that maximum difference can be positive or negative with respect to the order of the subtraction between datasets." What does this mean?

**L223: Changed to: "Note that the maximum difference can be either positive or negative, thus retaining the sign of RTTOV-LBL rather than just reporting the absolute difference between them."**

L153: It would help the reader to describe the figure (this applies to the other figures too, i.e., Figure 4, etc). i.e., "Figure 3 shows the differences in the (a) average, (b) standard deviation (STD) and (c) maximum values in the TOA Brightness temperatures (BT) simulations using the training profiles (red lines) and the independent profiles (blue lines) for all IRIS-D channels".

**L227: Reworded to - "Figure 3 shows the differences in the (a) average (AVG), (b) standard deviation (STD) and (c) maximum values (MAX) in the TOA Brightness Temperature (BT) simulations between RTTOV and LBLRTM for all IRIS-D channels."**

L155: What is for example the instrument channel errors in the assimilation systems? to put these differences in that context?

**L423: Instrument channel errors are highly variable and often several kelvin, so much higher than the validation statistics we have presented. The following website provides a variety of different assimilation statistics for three major weather centres -https://nwp-saf.eumetsat.int/site/monitoring/nrt-monitoring/. Most of the instruments we discuss are no longer in operation, however, SSMI/S is. We have added some discussion of this in the conclusion as follows (L425)- "Even with these increased performance errors produced by the larger dataset in the water vapour channels, these values are still much smaller than the instrument errors that assimilation systems have to deal with. For example, the mean and standard deviation of differences between observations and forecasted brightness temperatures are between of order 0.5--1.5 K for SSMI/S channels 9--11 still in operation on NOAA-17."**

L156: the CO2 band differences. I'm not sure if there are differences in the CO2 profiles (see questions above)

**The difference is not explain by the CO2 profiles as they are the same on both RT models but it is explain by the RTTOV error of the atmospheric transmittance parameterization on the temperature profiles in the CO2 band.**

L158: This contradicts what it says above "The differences between the two datasets are more evident in the CO2 band (between 600 and 800 cm−1) and in the ozone band (near 1000 cm−1)"

**The difference in both bands are explained by the temperature profiles and for the ozone band by the different variability in the ozone profiles in both profiles dataset.**

L174: What is happening at channels 14.95 µm and 22.91 µm

**Thank you very much for pointing that. We did not find a clear explanation on what happens at these two absorbing channels. In order to go further it would be useful to compare the weighting functions of these channels within the 83 versus the 25,000 profiles.**

L180: I guess SIRS-A channels are not shown? Except a figure is described? Which one?

**L251: You are right, the figure related to the SIRS-A results is not shown. However, we added in the text the results related to the SIRS-A. We modified the text to be more clear that we are talking about SIRS-A results (the figures of statistics are not shown for this instrument).**

L270: Some of this info was already stated in the methodology.

**If you are mean Zeeman effect it was mentioned once in the paper.**

Spatial variation of bias from the independent dataset

L285: What about the microwave? Otherwise these lines could go in the following paragraph with Figure 8. Perhaps it is illustrative to mark these channels in Figure 3. Why did you choose these channels?

**L322: The paragraph was rearranged to make it more logical as suggested: "For the infrared, three IRIS-D channels, two SIRS-B and one MRIR channels are shown. The three IRIS-D channels have a corresponding similar channel on another instrument (two in SIRS-B and one in MRIR) to test the robustness of the results. These comprise one surface channel (centred at 899 cm-1) one temperature (CO2) channel (centred at 679 cm-1) and one water vapour channel (centred at 1510 cm-1). " The 3 IRIS channels were identified by a star in the Figure 3. We used the Collard (2007) channel selection applied to IASI to guide our first selection in the IRIS-D. In the end, we selected some channels which there was a correspondent channel in two IR sensors explored during the project.**

Figure 8: why does IRID-D have a spatial variation in the bias, while the similar channel onboard SIRS-B doesn't? Is it only the instrument predictors that change? What about the bias? because figure 8 is BT-BT.

**L342: As mentioned in the paper "The reason for these differences is not entirely clear, but as the only difference between simulations of equivalent channels for both instruments are the bandwidths, with SIRS-B channels around a factor of 10 wider than IRIS-D, this is likely to be the cause.". Figures 8 and 9 are BT-BT.**

L295: It would aid the reader, a bit more guidance. Perhaps name the 899cm-1 channel before the 679cm-1 since Figure 8 was about the 899cm-1. Jumping to the latitudinal distribution of the 679cm-1 would help if information was given about the spatial distribution of this channel first. Not necessarily a figure like figure 8, but a description (?).

**L332: We rearranged the text to help the reader guidance as recommended. The new text is: "The figure clearly shows that the bias has latitudinal behaviour. The channel centred at 899 cm-1 tends to be negative in the equatorial region in the SIRS-B sensor (blue circles), whereas the corresponding channel in IRIS-D has a bias closer to zero, or slightly positive, in the equatorial region. To investigate this, we calculated the correlation between the mean bias and the integrated water vapour content (IWVC) that is provided for each of the 25,000 profiles. The correlation coefficient is moderate in the different sensors, however it is positive for the IRIS-D channel (0.48) and negative for SIRS-B (-0.47). , mainly for channel centred at 679 cm-1 (black circles). The channel centred at 679 cm-1 (black circles) presents a higher positive bias in all regions and the values are larger in the polar regions and there is an increase of the differences from the extratropical regions to the equatorial region, which is more evident in SIRS-B. There is no correlation (0.017) between mean bias and the IWVC for the channel 679 cm-1 of IRIS-D and the correlation is 0.40 for same channel of SIRS-B. The reason for these differences is not entirely clear, but as the only difference between simulations of equivalent channels for both instruments are the bandwidths, with SIRS-B channels around a factor of 10 wider than IRIS-D, this is likely to be the cause."**

L300. Please mention this is 'not shown'

**L343: We mention the figure is not shown in the text. "The spatial variability of SIRS-A channels (figure not shown) is similar the ones showed for SIRS-B channels."**

L301. Do you mean the latitudinal mean?

**L343: No, we mean spatial variability of SIRS-A, which can be compared with the spatial variability of SIRS-B represented in the Figures 8b and 9b.**

L306: Here you mention channels at 1051 cm-1 but in lines 288 these were not mentioned

**L345: There is a mistake in the typo of the channel wavenumber. The corrected channels are 1510.10 cm-1 from IRIS-D and 1510.03 cm-1 from MRIR. The modification were made in lines L346 and L347.**

Figure 10: maybe it helps to have smaller markers?

**We changed the size of the markers in the Figure 10.**

L314: Is the correlation between the total bias or the latitudinal bias? It is difficult to say that the spatial variability is possibly related to the content of water vapor, but later show no correlations. Do you have a similar figure to Figure 11b?

**The correlation is between the total bias and IWVC. We don't have a similar figure.**

L321: Its hard to see from the way the figure is plot which are the -2K bias results.

**L358: The nearly -2K biases are shown in scatter points on the left of the figure around the equatorial region. We modified the text: "The distribution of the bias around 0 K is reasonably symmetrical but there are a few profiles with very negative biases around the equatorial regions, up to a maximum bias of nearly -2 K."**

L324: Does this mean that the simulations included cloudy profiles but with no cloud particles? Did you evaluate screening those profiles from the statistics? What percentage of cloudy profiles were included?

**Yes, all simulations are performed in the clear-sky as there is no treatment of cloud in the LBL code even though there are clouds in some profiles. We did not evaluate screening those profiles from the statistics. As we didn't compute the statistics of cloud profiles the percentage of cloudy profiles were not computed.**

Profiles associated with maximum bias

The authors show the profile responsible for the maximum bias in each channel. Did you see how these profiles impacted the other channels? Are they also responsible for outlier behaviour?

**Some of the profiles responsible for outlier behaviour in each channel are the same profile for multiple channels. For example, the dark blue line in Figures 15--17 is the maximum outlier profile for at least 10 of the channels shown and 25 channels in total, this is already discussed in the text.**

L342: It is hard to read when in the text you use channel numbers but in the figures the channel frequencies.

**The text has been modified in different places (L382 and L387, for example) to make the identification of channels clearer.**

Why not conduct a similar analysis with the IR frequencies explored in the previous section?

**A similar analysis were conducted with the IR instruments, however there was not possible to find relation between the maximum bias values and profiles with different characteristics.**

Conclusions

L371. I am not sure I understand what you mean by 'This confirms that it is acceptable to validate the RTTOV coefficients using the same profiles used to generate the coefficients.'

**L408 and L428: We reworded to - "The results for the infrared sensors showed that the statistics for the independent profile dataset (25,000 profiles) are similar to those found when using the 83 training profiles, indicating the performance of RTTOV is robust against both datasets." Later on we state (L428) "Even though this study is restricted to historical sensors, the majority of which are no longer in operation, it confirms that the validation statistics for the 83 profile dataset are adequate to represent the overall biases for a range of different instruments.", which better explains the point - that we present validation statistics using the same profile set we create the coefficients with, and this is shown to be acceptable by this study.**

L390: The potential use of predictors for bias correction procedures is very interesting. It would be very valuable that at the heart of the discussion, the predictors were to be described in the introduction. In this analysis, only the zenith angle predictors are analyzed in a way for the microwave channels. The water vapor ones are sort of discussed too. However, wouldn't iwc and clw etc impact cloudy simulations? A robust discussion of the predictors would be good to have as a reference.

**L200: To clarify all simulations are performed in the clear-sky as there is no treatment of cloud in the LBL code. This is now made explicit in the RTTOV radiative transfer simulations ", and are all clear-sky in line with the line-by-line models which do not include any treatment of cloud or ice." As suggested we have included some more discussion of the predictors used in the RTTOV setup section (L206): "Version 7 predictors were introduced at the release of RTTOV-7, and are a development of those used in RTIASI. There are 10 predictors specified for the mixed gases (e.g. O2 and N2), 15 for water vapour and 11 for ozone, and they are functions of satellite zenith angle, temperature, water vapour and ozone mixing ratio, see Saunders and Rayer (2002), Table 1 for full details".**

General comments

• Would it be relevant to conduct a similar analysis for jacobian calculations?

**A similar analysis for Jacobian calculations could be interesting, see for example a recent comparison of RTTOV and LBL Jacobians in two AMSU-A channels is shown in Figure 5 of Saunders et al. (2018), where the Jacobians are shown to be quite similar. It is beyond the scope of the current study, however, as calculating LBL Jacobians is not straightforward.**

• Regarding surface emissivity. Are all radiative transfer models treating the surface in the same way? Would it be relevant to run the analysis for lower emissivities in the microwaves? thinking about the oceans for examples: most relevant for global assimilation systems.

**L218: All models are using an emissivity of 1, this is imposed by the LBL models having no treatment of reflectance. We make this explicit in the RTTOV radiative transfer simulation section (L218): "All calculations are performed using an emissivity value of 1, which is limited by the line-by-line models that simulate strictly up-welling radiation and do not calculate reflectance."**

• Regarding the RTTOV coefficient files. Were these calculated using the same absorption models as the ones used in this study? Because if there were calculated with different models, the spectroscopy would also be something to consider in the differences, right?

**The RTTOV coefficients were calculated using the same absorption models. They were calculated using the same spectroscopy.**

Tables and Figures

• Table 1: for completeness include the channels, or otherwise change "Channels" to "Number of channels"

**We replace "Channels" by "Number of channels" in the table 1.**

• Figure 1 legend please rephrase.

**The legend was rephrased as follows: a) 25,000 profiles of temperature [K] of independent dataset, b) 83 training profiles of temperature [K] c) 25,0000 profiles of water vapour [ppmv] of independent dataset, d) water vapour profiles of 83 training dataset. e) and f) ozone [ppmv] profiles of independent dataset and training dataset, respectively. The mean profile is shown for 25,000 (black lines) profiles and training profiles (red lines). The maximum (blue lines) and minimum (yellow lines) profiles of 83 training profiles are also shown.**

• Figure 2: legend and plot titles are not self explanatory.

**We unified the figures. We decided to remove the figures titles and explain better the figures in the caption. The new caption is: "a) spatial and b)vertical distribution of 87 ozone profiles from ozone subset which present double peak of ozone in the vertical distribution. These profiles have a second maximum of ozone quantity**

**(higher than 3 ppmv) above 0.9hPa. The mean ozone profile is represented by blue line."**

• Figure 3: Please put y-axis labels (mean, std, max) and in the legend training vs. independent profiles.

**We unified the figures. We decided to remove the figures titles and explain better the figures in the caption.**

• In general all figures should have self explanatory titles that have the same format. For example, in Figure 10 it says '7 54' but in Figure 9 '7pred54'. Figure 14 for example is missing a, b and c. subplot c doesn't have a title. This applies to many figures. Please also unify font size in all figures.

**We unified the figures. We decided to remove the figures titles and explain better the figures in the caption.**
* * *
**Reviewer 2:**

Review of "Global evaluation of RTTOV coefficients for early satellite sensors"

Summary: The authors evaluated RTTOV version 7 coefficients for nine historical sensors using a dataset of 25,000 profiles along with the standard 83 profile training dataset. RTTOV output was compared to output from a line by line model for each sensor to perform the evaluation. Average biases as well as the spatial distribution of RTTOV biases are presented for each sensor. For the most part, the paper is well written and logically organized, I have only a few points to raise, mainly to improve the clarity of the information presented. I therefore recommend minor revisions.

**We thank Reviewer 2 for his/her comments, questions and remarks, which we hope helped to improve the quality of the original manuscript. Please find below our responses to your comments. Reviewer 2's comments are in** normal **font, our answers are written with bold font.**

Comments:

Line 56 "every instruments": instrument

**L63: We modified as suggested. "...though not every molecule is included for every instrument, depending on its spectral absorption coverage."**

Figure 1: I'm not colorblind, so I can't confirm if it is in fact difficult to see, but please consider that having a red line and a green line, especially right next to each other, may not be the most accessible color choice.

**We modified figure 1. We replaced the green lines with black lines.**

Section 3.1: I would suggest switching the order of sections 3.1 and 3.2 since the profiles that failed the LBL are excluded from the RTTOV simulations, but the LBL model setup (and the failing profiles) hasn't been discussed yet, which makes it feel a little out of order.

**We switched the order of sections 3.1 and 3.2 as suggested. And we also added a section "Satellite sensors" (L99). The sections 3.1 and 3.2 became section 4.2 (L199) and section 4.1 (L161).**

Line 119 "55 profiles failed": Were these profiles also not used for the AMSUTRAN simulations?

**These were profiles that fell outside of some internal checks that LBLRTM imposed on profile limits. Using a more recent version of LBLRTM revealed that these checks had been modified and the same profiles no longer failed. AMSUTRAN processed all profiles without any failures. As it was only a few profiles out of 25,000 their omission will likely have had no effect on the resulting statistics, so we suggest we remove the comment to avoid confusion.**

Line 190 "Other two": Replacing with 'Two other' would make more sense.

**L262: We modified as suggested. "Two other sensors were also analysed."**

Line 190: I understand this is only one short paragraph, but it's a little strange that these sensors are in the SIRS section, rather than their own.

**L261: We decided not show the HRIR and MRIR statistics figures because they don't provide additional information. However, we want to add some information about these results. We added a unique subsection "5.1.3 HRIR and MRIR".**

Line 194 "the other two sensors": So far you have discussed four other sensors, which two are you referring to here?

**L263: We are referring IRIS-D and SIRS. We changed it in the text. "The statistics of MRIR is similar to the SIRS and IRIS-D, except in the channel centred at 17.06 µm which presents the highest mean differences, standard deviation and maximum value (figure not shown)."**

Line 206 "satellite zenith angle": Please include the SZA acronym definition here.

**L213: We included the acronym in the first time it appears in the text. "For the microwave simulations, the standard six satellite zenith angles (SZA) that vary between 0.0º to 63.6º were used, which equates to secant values of: 1.0, 1.25, 1.5, 1.75, 2.0 and 2.25. "**

Section 4.1.7: Is there a problem with channel 8?

**L156: The statement "Channel 8 at 150 GHz is a window channel." was added for completeness.**

Line 276 "channels 7-11 around 183.31 GHz": Line 265 seems to imply that channels 17-18 are the ones around 183 GHz.

**L312: These errors have been corrected. "The water vapour channels 9--11 around 183.31 GHz show the biggest differences between RTTOV and AMSUTRAN, and between the training profile set and the 25,000 profile set, as expected based on corresponding channels on SSM/T-2."**

Line 276 "channels 7-11 around 183.31 GHz": Do you mean channels 9-11?

**L312: These errors have been corrected. "The water vapour channels 9--11 around 183.31 GHz show the biggest differences between RTTOV and AMSUTRAN, and between the training profile set and the 25,000 profile set, as expected based on corresponding channels on SSM/T-2."**

Line 285 "2 IRIS-D channels ... The three IRIS-D channels": Please check the description of the number of channels on this line, do you mean three IRIS-D channels and two SIRS-B? If not, why is the corresponding CO2 channel for SIRS-B not shown?

**L322: We corrected it in the text. "For the infrared, three IRIS-D channels, two SIRS-B and one MRIR channels are shown."**

Line 294 "Figures 9a and 9b ... respectively": This sentence is confusing and makes it sound like only the 679 GHz channel from IRIS-D and the 899 GHz channel from SIRS-B is shown, especially when paired with the description at the top of the section (see previous comment)

**L331: We rearranged the sentence as follows: "Figure 9a shows the latitudinal distribution of the channels centred at 679 cm-1 and 899 cm-1 from the IRIS-D and Fig. 9b the same two channels from SIRS-B."**

Figure 9: Figure 8 and Figure 9b show essentially the same information, it seems slightly redundant to show both, especially since the discussion is also repeated (Lines 291-293 and lines 298-299)

**We think that both figures are informative, Figure 8 shows a global view of the differences, whereas Figure 9b specifies that the parametrization error due to the training over 83 profiles may be modelized as function of the latitude, that can be of interest for bias correction or profile retrieval methods for example. Furthermore, few channels can be compared within the same plot. We will keep these two figures but we remove lines 298-299 to avoid repetition.**

Figures 8-10: Is there a reason these 3 figures are not presented in the same way? Why is Figure 9 not similar to Figures 8 and 10 but for the 679 GHz channel? Alternatively, the three figures could be combined into one, similar to Figure 9 but with 3 panels.

**We made a lot of figures for all channels and all instruments for both ways and selected the most representative and the most interesting results from all these figures.**

Figures 14-18: The layout of these figures makes them quite small and difficult to read when formatted by the journal. Consider rearranging these slightly, perhaps sharing the y-axis (removing the repeated pressure label and tick labels) or trying an arrangement with 2 rows (it may also help to move the legend outside the plot area as the 4th "panel")

**These figures have been reformatted to make them clearer.**

Figures 15-18: I don't think it's necessary to include the legend on each panel, this may help with the readability issues.

**These figures have been reformatted to make them clearer.**
* * *
**Reviewer 3:**

Review of the manuscript "Global evaluation of RTTOV coefficients for early satellites sensors"

**We thank Reviewer 3 for his/her comments, questions and remarks, which we hope helped to improve the quality of the original manuscript. Please find below our responses to your comments. Reviewer 3's comments are in** normal **font, our answers are written with bold font.**

General comments

1. This manuscript presents important results to quantify the performance of the radiative transfer model, RTTOV, that is used by numerical weather prediction and climate reanalysis centers, such as ECMWF and JMA, for example.

2. This study considers specifically early satellite sensors. Given this focus, it would be important to verify the performance of the RTTOV model to handle known variations between environmental conditions at the time of early satellite sensors and present-day conditions. These variations may have affected the temperature, humidity, and ozone, all well discussed in the paper.

3. However, further environmental variations have also affected, with significant net fractional changes, several absorbers' concentrations. One would hence expect that a particular attention be given in such a study to the performance of the (RTTOV) model to handle (known) changes in $CO_2$ (as well as $CH_4$ and $N_2O$), to only name these species. Carbon dioxide is probably the most important in this respect because its 15 micron absorption line is used for infrared temperature 'profiling' (although in practice all

wavenumbers contribute to extract information, in a data assimilation system), and oxygen is similarly interesting if one considers that it has some non-negligible variability (see reference below). This aspect of variability in the absorber amounts remains a bit hidden in my opinion (recognizing it is discussed adequately for humidity and ozone) and would deserve to be more exposed. Including this variability may increase significantly the magnitudes of differences between training and independent set.

**We thanks the reviewer for this comment and agree with it. However, the effect of species such as CO2, CH4 or N2O on simulated observations was not possible to be studied in the frame of our work as we used a large dataset of atmospheric profiles coming from NWP model. We focused our study on the NWP application of reanalysis where water vapour and ozone are the most important gaseous species in both IR and MW spectral domains.**

4. There are other species that have seen substantial changes since the 1970s, e.g., ozone depleting substances such as CFCs. For these ones in particular, one would expect to see an impact on the performance of RTTOV to model several channels of the IRIS instrument in 1970.

**We agree on that but the CFCs or other species are beyond the scope of the paper. RTTOV does not allow for varying all species due to the fast atmospheric transmittance model. The performance of RTTOV in the ozone band near 1000 cm-1 is given in the paper.**

5. A related point is that, similarly to controlling the performance of RTTOV to reproduce past variations, given assumed environmental trends, one could imagine that a similar study be of interest to several operators such as NWP centers to predict at what point, in the future, the present rates of increases in CO2, CH4, and N2O amounts may yield to significant radiative transfer modelling errors, if their variability is not sufficiently represented in the training profiles.

**Yes, we agree on that too and this was done in the training profile of RTTOV for version 12 were they extend the variability of the training profile for CO2, CH4 and N2O to take into account the increase over the last decades. More information is given** **here** **https://nwp-saf.eumetsat.int/site/download/documentation/rtm/docs_rttov12/ rttov12_svr.pdf**

Detailed comments

1. L16: could one make a link here with 'machine learning'?

**L17: We added the sentence "RTTOV is also more and more used to train machine-learning based approach for simulating satellite observations (e.g., Scheck, 2021)."**

2. L 19: "small": this adjective may be removed – or else, quantified.

**L20: This paragraph was rearranged "The accuracy of the RTTOV transmittance parameterisation was firstly analysed by Saunders et al.,2007 that shown an overall agreement within 0.05 K between different RTM and LBL models, except for certain spectral regions. The evaluation was based on a subset of 49 atmospheric profiles selected from a large atmospheric profiles dataset of the ECMWF ERA-40 database."**

3. L 30: "to be released": did you mean to write "to enter production"

**L34: We changed it as suggested. "The aims were to retrieve/rescue and reprocess historical infrared and microwave (MW) meteorological satellite observations from the 1970's and 1980's, primarily for inclusion in the ECMWF ERA6 reanalysis, the follow-on to ERA5 (Hersbach et al., 2020), which is due to enter production in 2024."**

4. L 31: "all satellite observations": add the word "radiance"

**L37: We changed it as suggested. " In these reanalyses, RTTOV is employed to simulate all satellite radiance observations."**

5. L 35: "a more robust": it is not just about robustness here. A necessary validation is indeed to verify that the application of RTTOV, to the training profiles, does work, as intended. The validation that is presented here is "additional", in my opinion (i.e., it does not replace the step of necessary validation with the training profiles).

**L42: Changed to - "A further, and more rigorous validation can be obtained by employing a larger independent profile dataset. "**

6. What is the IFS version used to produce the 25,000 profiles?

**L72: Version Cycle 40r1 of the Integrated Forecasting System (IFS), Eresma and McNally (2014). We added this information in the text as follow (L72): "The dataset is selected from the short-range IFS (cycle 40r1) forecast over one year and is available from the NWPSAF website."**

7. How relevant or useful may it be to consider (in potential future studies) using atmospheric profiles from a completely different source, other than ECMWF, for performance assessment?

**This could be interesting as both profile sets come from ECMWF models, maybe using high resolution radiosondes directly or something, in the future. We doubt they will produce very different results, however.**

8. L 97-98: What is the relevance of deriving coefficients at 400 ppmv CO2 for early sensors, when this concentration is approximately 25% larger than actual concentrations at the time of the early sensors?

**The value of 400 ppm for CO2 is too high for early satellite observations, a better value would be 320 ppm, but in this study we did not used observations but we used NWP atmospheric profiles for comparison where the CO2 amount is not**

**varying. When a gaseous specie does not vary in the simulation, RTTOV used the mean profile value for all simulation. We think that the value of 400 ppm does not change the overall results of our paper. Furthermore for reanalyses, the CO2 amount is varying and in that case, the RTTOV coefficients are made such as it allow for varying CO2.**

9. Is it indeed the case that there are no differences in this study between the absorber amount concentrations (except for water vapour and ozone) between the training profile set and the independent profile set? If so, then the performance of RTTOV will necessarily be over-confident, because important sources of variability are neglected (unless proven or otherwise).

**Yes, it is true that for channels that are more sensitive to other species than water or ozone, the results of this study is over confident but these two species are the most important for NWP. However a dedicated study on other species would be interesting to do in the future.**

10. L 92: about the profiles that failed LbL calculations: how many profiles does this represent?

**These were profiles that fell outside of some internal checks that LBLRTM imposed on profile limits. Using a more recent version of LBLRTM revealed that these checks had been modified and the same profiles no longer failed. AMSUTRAN processed all profiles without any failures. As it was only a few profiles out of 25,000 their omission will likely have had no effect on the resulting statistics, so we suggest we remove the comment to avoid confusion. In the simulations with 101 levels, 40 profiles failed (0.16%) and the simulations with 54 levels, 55 profiles failed (0.22%).**

11. In each subsection 3.2.X, it could be useful to remind the reader which absorber(s) are sensed by the instrument (when this information is not already present).

**L99: The "section 3.1 - Satellite Sensors" was created. This section presents a brief description of the sensors used in the present study. The objectives of these sensors were also presented.**

12. On the importance of the ISRF: is there any obvious relationship observed between the channel bandwidth and the model performance found?

**There can be some performance differences depending on the part of the spectrum considered and the bandwidth of the channel, for example if there are many lines within the bandwidth. This is revealed by looking at the difference between the first two validation figures shown on the NWPSAF website, see for example https://nwp-saf.eumetsat.int/downloads/rtcoef_rttov13/visir_lbl_comp/ lbl_comp_rtcoef_noaa_19_hirs_o3_v13pred_54L.html , where the first plot is the**

**averaged radiance over the channel (a LBL calculation) and the second is the RTTOV radiance. There are negligible differences in MW channels.**

13. Conclusions: one aspect that stands out in the results is that best agreement is obtained for channels when the absorber amount is (if I understood well) kept identical, between training and independent profiles, i.e., oxygen and CO2 channels in particular. Can you confirm?

**Yes, but see other discussion on the limitations of this study.**

14. Conclusions: the sentence " This confirms that it is acceptable to validate the RTTOV coefficients using the same profiles used to generate the coefficients " is a bit too far reaching.

**L408: This has been changed now to - "The results for the infrared sensors showed that the statistics for the independent profile dataset (25,000 profiles) are similar to those found when using the 83 training profiles, indicating the performance of RTTOV is robust against both datasets." Later on we state (L428), "Even though this study is restricted to historical sensors, the majority of which are no longer in operation, it confirms that the validation statistics for the 83 profile dataset are adequate to represent the overall biases for a range of different instruments."**

Editorial comments

1. Introduction: maybe a sentence explaining how a 'fast' radiative transfer model is constructed, before L 16, would help. (e.g. based on LbL calculations, use of predictors…)

**We think the way round we have it is more logical, to explain the overall use, then the detail of how it constructed. We also go into more detail in the RTTOV setup about predictors now.**

2. L 42: profile -> profiler

**L49: It is was modified as follows: The modification were made as suggested "... Special Sensor Microwave - Humidity (SSM/T-2) , Scanning Multichannel Microwave Radiometer (SMMR) and Special Sensor Microwave Imager/Sounder (SSMI/S), Table 1."**

3. L 43: Sounde -> Sounder

**L50: The modification were made as suggested "... Special Sensor Microwave - Humidity (SSM/T-2) , Scanning Multichannel Microwave Radiometer (SMMR) and Special Sensor Microwave Imager/Sounder (SSMI/S), Table 1."**

4. L 65-66: could you rephrase the sentence "Each of the 5,000 profile subsets..."

**L72: The 5 subsets of 5,000 profiles represent maximum variability of one of five different variables: temperature (t), specific humidity (q), ozone (o3), cloud condensate (ccol) or precipitation (rcol).**

5. L 78: use either dot or the colon as the decimal separator (using both may be confusing – and given the use of the colon as thousand separator throughout the paper, e.g. "25,000" for the number of profiles, I would recommend using the dot as decimal separator).

**We changed to colon.**

6. Figure 19 caption, typo: "are some are" -> "and some are"

**It was changed in the Fig. 19 caption. "There are less than 50 points and some are associated with multiple channels."**

7. Throughout the paper, unless you refer to a 'historic' event, maybe the adjective use is more often 'historical' than 'historic'.

**We changed historic by historical in the paper.**

References of potential interest

1. Shi, P., Chen, Y., Zhang, G.et al.Factors contributing to spatial–temporal variations of observed oxygen concentration over the Qinghai-Tibetan Plateau.Sci Rep11, 17338 (2021). https://doi.org/10.1038/s41598-021-96741-6

**We did not added this reference because the oxygen is not a variable gas used in this study.**

---

## Author Response (AR2)

We thank Reviewer 2 for his/her comments, questions and remarks, which we hope helped to improve the quality of the original manuscript. Please find below our responses to your comments. Reviewer 2's comments are in normal font, our answers are written with bold font.

1. Line 127 "used to temperature": used for temperature
**L108: We agree with you. We changed the text as suggested.**

2. Line 128 "used to surface": used for surface
**L109: We agree with you. We changed the text as suggested.**

3. Line 134 "flown on 3": flown on Nimbus 3
**L116: We agree with you. We changed the text as suggested.**

4. Line 134 "present two": had two
**L116: We agree with you. We changed the text as suggested.**

5. Line 159 "the remaining channel vertically": the remaining channel being vertically
**L137: We agree with you. We changed the text as suggested.**

6. Line 223 "too": please remove
**L198: We agree with you. We changed the text as suggested.**

7. Line 250 "thus retaining the": thus we retain the
**L223: We agree with you. We changed the text as suggested.**

8. Line 379 "so will never utilise the full range of SZA calculated for the standard RTTOV coefficients": therefore the the full range of SZA calculated for the standard RTTOV coefficients will never be used
**L307: We agree with you. We changed the text as suggested.**

9. Line 412 "moderate in the different sensors": moderate for both sensors
**L336: We agree with you. We changed the text as suggested.**

10. Figure 3 caption "3 starts": 3 stars
**We corrected it.**